# HoTPP Benchmark: Are We Good at the Long Horizon Events Forecasting?

## Abstract

Accurately forecasting multiple future events within a given time horizon is crucial for finance, retail, social networks, and healthcare applications. Event timing and labels are typically modeled using Marked Temporal Point Processes (MTPP), with evaluations often focused on next-event prediction quality. While some studies have extended evaluations to a fixed number of future events, we demonstrate that this approach leads to inaccuracies in handling false positives and false negatives. To address these issues, we propose a novel evaluation method inspired by object detection techniques from computer vision. Specifically, we introduce Temporal mean Average Precision (T-mAP), a temporal variant of mAP, which overcomes the limitations of existing long-horizon evaluation metrics. Our extensive experiments demonstrate that models with strong next-event prediction accuracy can yield poor long-horizon forecasts and vice versa, indicating that specialized methods are needed for each task. To support further research, we release HoTPP[1], the first benchmark designed explicitly for evaluating long-horizon MTPP predictions. HoTPP includes large-scale datasets with up to 43 million events and provides optimized procedures for both autoregressive and parallel inference, paving the way for future advancements in the field.

## 1 Introduction

The world is full of events. Internet activity, e-commerce transactions, retail operations, clinical visits, and numerous other aspects of our lives generate vast amounts of data in the form of timestamps and related information. In the era of AI, it is crucial to develop methods capable of handling these complex data streams. We refer to this type of data as Event Sequences (ESs). Event sequences differ fundamentally from other data types. Unlike tabular data (Wang & Sun, 2022), ESs include timestamps and possess an inherent order. In contrast to time series data (Lim & Zohren, 2021), ESs are characterized by irregular time intervals and additional data fields. These differences necessitate the development of specialized models and evaluation practices.

Sequence modeling is the primary task in the Event Sequences (ESs) domain. In its simplest form, each event is defined by its type and occurrence time, a framework commonly known as Marked Temporal Point Processes (MTPP) (Rizoiu et al., 2017). Some studies extend MTPP to incorporate additional data fields and model complex dependencies between them (McDermott et al., 2024). However, the majority of MTPP approaches, as well as their evaluation pipelines, primarily focus on predicting the next event based on historical data.

In practice, a common question arises: what events will occur, and when, within a specific time horizon? Forecasting multiple future events presents unique challenges that differ from traditional next-event prediction tasks. For instance, autoregressive event sequence prediction involves applying the model to its own potentially erroneous predictions. However, the challenges of autoregressive prediction in the context of MTPP have not been thoroughly explored. Another difficulty lies in evaluation. Methods like Dynamic Time Warping (DTW) are typically unsuitable for event sequences due to strict ordering constraints (Su & Hua, 2017). Some works have applied Optimal Transport Distance (OTD), a variant of the Wasserstein distance, to compare sequences of predefined lengths. Still, the limitations of this metric have not been previously considered.

---

[1] https://github.com/anonymous-10647849/hotpp-benchmark-submission

Predict events on the horizon

Model

Temporal mAP

Timestamps and labels distributions

Prefix

Ground truth

Event sequence

Figure 1: Temporal mAP (T-mAP) evaluation pipeline. Unlike previous methods, T-mAP evaluates sequences of variable lengths within the prediction horizon. It further improves performance assessment by analyzing label distributions rather than relying on fixed predictions.

In this work, we provide the first in-depth analysis of models and metrics for long-horizon event forecasting, establishing a rigorous evaluation framework and a baseline for MTPP studies. Our key contributions are as follows:

1. We demonstrate that widely used evaluation methods for MTPPs often overlook critical aspects of model performance. We demonstrate that simple rule-based baselines can sometimes outperform popular deep learning methods when evaluated using OTD.

2. We introduce Temporal mean Average Precision (T-mAP), a novel evaluation metric inspired by best practices in computer vision. T-mAP evaluates variable length sequences within a specified time horizon, as illustrated in Figure 1. Unlike previous approaches, T-mAP accurately accounts for false positives and false negatives while being invariant to linear calibration. Additionally, we address a theoretical gap in computer vision by proving the correctness of the T-mAP computation algorithm.

3. Using our established methodology, we demonstrate that high next-event prediction accuracy does not necessarily translate into high-quality long-horizon forecasts; in many cases, our experiments show the opposite. This highlights the necessity of developing specialized models for the long-horizon prediction task.

4. We release HoTPP, a new open-source benchmark designed to facilitate long-horizon event sequence prediction research. HoTPP brings together datasets and methods from various domains, including financial transactions, social networks, healthcare, and recommender systems, greatly expanding the diversity and scale of data compared to prior benchmarks. Additionally, we offer an efficient inference algorithm necessary for large-scale evaluations.

## 2 RELATED WORK

**Marked Temporal Point Processes.** A Marked Temporal Point Process (MTPP) is a stochastic process consisting of a sequence of pairs $(t_1, l_1), (t_2, l_2), \ldots$, where $t_1 < t_2 < \ldots$ represent event times and $l_i \in \{1, \ldots, L\}$ denote event type labels (Rizoiu et al., 2017). Common approaches to MTPP modeling focus on predicting the next event. A basic solution involves independently predicting the event time and type. A more advanced approach splits the original sequence into subsequences, one for each event type, and independently models each subsequence's timing. Depending on the time-step distribution, the process is called Poisson or Hawkes.

Over the last decade, the focus has shifted toward increasing model flexibility by applying neural architectures. Several works have employed classical RNNs (Du et al., 2016; Xiao et al., 2017; Omi et al., 2019) and transformers (Zuo et al., 2020; Zhang et al., 2020), while others have proposed architectures with continuous time (Mei & Eisner, 2017; Rubanova et al., 2019; Kuleshov et al., 2024). In this work, we evaluate MTPP models with both discrete and continuous time architectures. Unlike previous benchmarks, we also assess simple rule-based predictors and popular methods from related fields, including GPT-like prediction models for event sequences (McDermott et al., 2024; Padhi et al., 2021) and Next-K models from time series analysis (Lim & Zohren, 2021). For more details on MTPP modeling, refer to Appendix A.

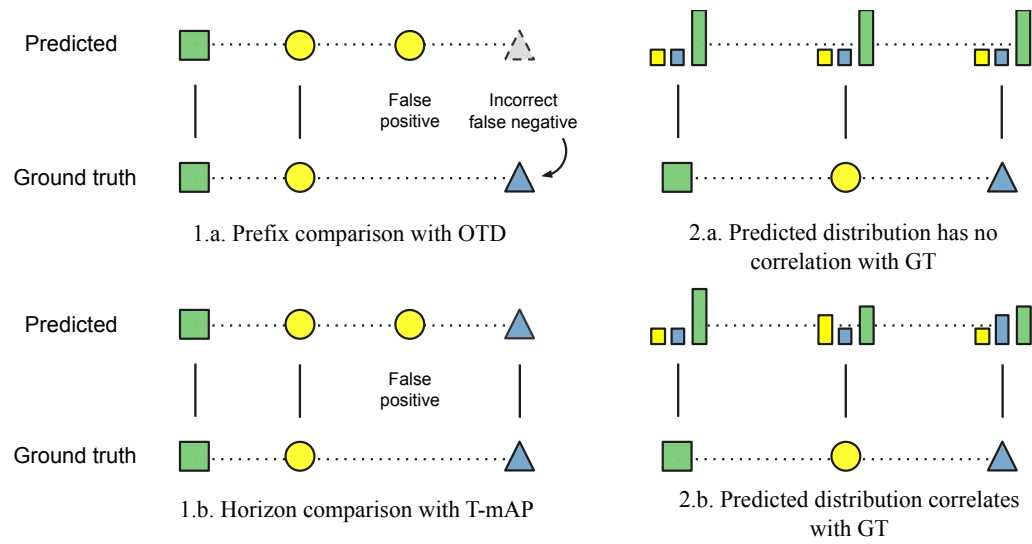

Figure 2: Comparison of OTD and T-mAP metrics. In example 1.a prefix evaluation of three events using OTD results in an incorrect, false negative. T-mAP addresses this issue by comparing all events within the time horizon, as shown in example 1.b. Additionally, OTD evaluates only the label with the maximum probability. From the OTD perspective, cases 2.a and 2.b have the same quality. In contrast, T-mAP evaluates the entire distribution, yielding a low score of 0.33 in case 2.a and a high score of 1 in case 2.b.

**MTPP Evaluation.** Previous benchmarks have primarily focused on medical data or traditional MTPP datasets. EventStream-GPT (McDermott et al., 2024) and TemporAI (Saveliev & van der Schaar, 2023) consider only medical data and do not implement methods from the MTPP field, despite their applicability. Early MTPP benchmarks, such as Tick (Bacry et al., 2017) and Py-Hawkes[2], implement classical machine learning approaches but exclude modern neural networks. While PoPPy (Xu, 2018) and EasyTPP (Xue et al., 2023) include neural methods, they do not consider rule-based and Next-K approaches. Furthermore, PoPPy does not evaluate long-horizon predictions at all. EasyTPP addresses this limitation to some extent by reporting the OTD metric, though it does not provide the corresponding evaluation code.

Previous work also highlighted the limitations of Dynamic Time Warping (DTW) for event sequence evaluation (Su & Hua, 2017), showing that DTW's strict ordering constraints are impractical and should be avoided. This work proposed an alternative, the Order-preserving Wasserstein Distance (OPW). In our study, we address the limitations of OTD, a variant of the Wasserstein Distance, and propose a new metric called T-mAP. Unlike OPW, T-mAP operates on timestamps rather than event indices and involves two hyperparameters, compared to three in OPW.

## 3 LIMITATIONS OF THE NEXT-EVENT AND OTD METRICS

MTPPs are typically evaluated based on the accuracy of next-event predictions, with time and type predictions assessed independently. The quality of type predictions is measured by the error rate, while time prediction error is evaluated using either Mean Absolute Error (MAE) or Root Mean Squared Error (RMSE). However, these metrics do not account for the model's ability to predict multiple future events. For example, in autoregressive models, the outputs are fed back as inputs for subsequent steps, which can lead to cumulative prediction errors. As a result, long-horizon evaluation metrics are necessary.

---

[2]https://github.com/slinderman/pyhawkes

Recent studies have advanced the measurement of horizon prediction quality by employing OTD (Mei et al., 2019). This metric is analogous to the edit distance between the predicted event sequence and the ground truth. Suppose there is a predicted sequence $S_p = \{(t_i^p, l_i^p)\}_{i=1}^{n_p}, 0 < t_1^p < t_2^p < \ldots$, and a ground truth sequence $S_{gt} = \{(t_j^{gt}, l_j^{gt})\}_{j=1}^{n_{gt}}, 0 < t_1^{gt} < t_2^{gt} < \ldots$. These two sequences form a bipartite graph $\mathcal{G}(S_p, S_{gt})$. The prediction is connected to the ground truth event if their types are equal. Denote $M(S_p, S_{gt})$ a set of all possible matches in the graph $\mathcal{G}(S_p, S_{gt})$. OTD finds the minimum cost among all possible matchings:

$$\text{OTD}(S_p, S_{gt}) = \min_{m \in M(S_p, S_{gt})} \left[ \sum_{(i,j) \in m} |t_i^p - t_j^{gt}| + C_{del} U_p(m) + C_{ins} U_{gt}(m) \right], \quad (1)$$

where $U_p(m)$ is the number of unmatched predictions in matching $m$, $U_{gt}(m)$ is the number of unmatched ground truth events, $C_{ins}$ is an insertion cost, and $C_{del}$ is a deletion cost. It is common to take $C_{ins} = C_{del}$ (Mei et al., 2019). It has been proven that OTD is a valid metric, as it is symmetric, equals zero for identical sequences, and satisfies the triangle inequality.

OTD is computed between fixed-size prefixes, but this approach limits metric flexibility in assessing models with imprecise time step predictions, as illustrated in Fig. 2.1. If the model predicts events too frequently, the first $K$ predicted events will correspond to the early part of the ground truth sequence, resulting in false negatives. This outcome is inaccurate because allowing the model to predict more events would alter the evaluation. Conversely, if the time step is too large, the first $K$ predicted events will extend far beyond the horizon of the first $K$ target events, leading to a significant number of false positives. This is undesirable as it prevents the inclusion of additional ground truth events in the evaluation. Therefore, evaluating a dynamic number of events is essential to align with the ground truth's time horizon properly.

MTPP evaluation metrics can be categorized into two groups: those considering event indices and those assessing time and label prediction quality independently of actual positions. Next-event metrics rely on ordering, even when events share identical timestamps, making the order meaningless. While OTD itself is invariant to order, the extraction of sequences for comparison is influenced by event indices. For example, when OTD compares length prefixes $K$, the algorithm must place $K$ correctly predicted events at the beginning of a sequence to minimize the OTD score. As a result, OTD, like next-event metrics, can depend on event ordering even when the order cannot be uniquely determined. This dependency is non-trivial and difficult to measure or control accurately.

Another limitation of the OTD metric is its inability to evaluate the full predicted distribution of labels, as shown in Figure 2.2. OTD considers only the labels with the highest probability, ignoring the complete distribution. This makes it dependent on model calibration and limits the ability to assess performance across a broader range of event types, such as long-tail predictions. However, models typically predict probabilities for all classes, allowing for a more comprehensive assessment. Therefore, we aim to develop an evaluation metric that accurately captures performance across common and rare classes.

## 4 TEMPORAL MAP: PREDICTION AS DETECTION

In this section, we introduce a novel metric, *Temporal mAP (T-mAP)*, which analyses all errors within a predefined time horizon, explicitly controls ordering, and is invariant to linear calibration. T-mAP is inspired by object detection methods from computer vision (Everingham et al., 2010), as illustrated in Fig. 1. Object detection aims to localize objects within an image and identify their types. In event sequences, we tackle a similar problem but consider the time dimension instead of horizontal and vertical axes. Unlike object detection, where objects have spatial size, each event in an MTTP is a point without a duration. Therefore, we replace the intersection-over-union (IoU) similarity between bounding boxes with the absolute time difference.

T-mAP incorporates concepts from OTD but addresses its limitations, as outlined in the previous section. Firstly, T-mAP evaluates label probabilities instead of relying on final predictions. Secondly, T-mAP restricts the prediction horizon rather than the number of events. These adjustments result in significant differences in formulation and computation, detailed below.

## 4.1 DEFINITION

T-mAP is parameterized by the horizon length $T$ and the maximum allowed time error $\delta$. T-mAP compares predicted and ground truth sequences within the interval $T$ from the last observed event. Consider a simplified scenario with a single event type $l$. Assume the model predicts timestamps and presence scores (logits or probabilities) for several future events: $S_p^l = \{(t_i^p, s_i^p)\}, 1 \leq i \leq n_p$. The corresponding ground truth sequence is $S_{gt}^l = \{t_j^{gt}\}, 1 \leq j \leq n_{gt}$. For simplicity, assume the sequences $S_p^l$ and $S_{gt}^l$ are filtered to include only events within the horizon $T$.

For any threshold value $h$, we can select a subset of the predicted sequence $S_p^l$ with scores exceeding the threshold:

$$S_>^l(h) = \{t_i^p : \exists(t_i^p, s_i^p) \in S_p^l, s_i^p > h\}. \tag{2}$$

By definition, a predicted event $i$ can be matched with a ground truth event $j$ iff $|t_i^p - t_j^{gt}| \leq \delta$, meaning the time difference between the predicted and ground truth events is less than or equal to $\delta$. T-mAP identifies the matching that maximizes precision and recall, i.e., matching with maximum cover $c$. The precision of this matching is calculated as $c/|S_>^l(h)|$ and the recall as $c/|S_{gt}^l|$. For any threshold $h$, we can count true positives (TP), false positives (FP), and false negatives (FN) across all predicted and ground truth sequences. Note that there are no true negatives, as the model cannot explicitly predict the absence of an event. Similar to binary classification, we can define a precision-recall curve by varying the threshold $h$ and then estimate the Average Precision (AP), the area under the precision-recall curve. Finally, T-mAP is defined as the average AP across all classes.

## 4.2 COMPUTATION

Finding the optimal matching independently for each threshold value $h$ is impractical; thus, we need a more efficient method to evaluate T-mAP. This section shows how to optimize T-mAP computation using an assignment problem solver, like the Jonker-Volgenant algorithm (Jonker & Volgenant, 1988). The resulting complexity of T-mAP computation is $\mathcal{O}(BN^3)$, where $B$ is the number of evaluated sequences and $N = \max(n_p, n_{gt})$ is the number of events within the horizon.

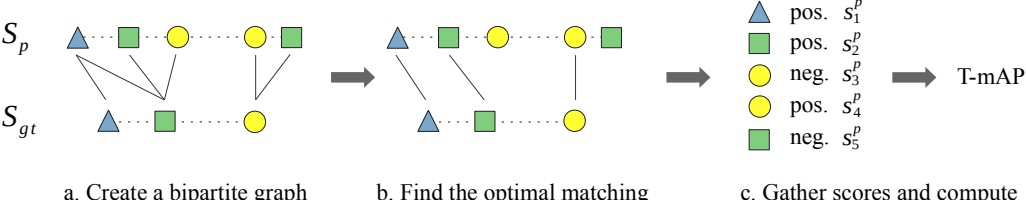

a. Create a bipartite graph      b. Find the optimal matching      c. Gather scores and compute

Figure 3: T-mAP computation pipeline.

For each pair of sequences $S_p^l$ and $S_{gt}^l$ we define a weighted bipartite graph $\mathcal{G}(S_p^l, S_{gt}^l)$ with $|S_p^l|$ vertices in the first and $|S_{gt}^l|$ vertices in the second part. For each pair of prediction $i$ and ground truth event $j$ with $|t_i^p - t_j^{gt}| \leq \delta$ we add an edge with weight $-s_i^p$, equal to negative logit of the target class, as shown in Fig. 3.a. Jonker-Volgenant algorithm finds the matching with the maximum number of edges in the graph, such that the resulting matching minimizes the total cost of selected edges, as shown in Fig. 3.b. We call this matching *optimal matching* and denote a set of all optimal matching possibilities as $M(\mathcal{G})$. For any threshold $h$, there is a subgraph $\mathcal{G}_h(S_>^l(h), S_{gt}^l)$ with events whose scores are greater than $h$. The following theorem holds:

**Theorem 4.1.** *For any threshold $h$ there exists an optimal matching in the graph $\mathcal{G}_h$, that is a subset of an optimal matching in the full graph $\mathcal{G}$:*

$$\forall h \forall m \in M(\mathcal{G}) \exists m_h \in M(\mathcal{G}_h) : m_h \subset m. \tag{3}$$

According to this theorem, we can compute the matching for the prediction $S_{gt}^l$ and subsequently reuse it for all thresholds $h$ and subsequences $S_>^l(h)$ to construct a precision-recall curve for the entire dataset, as shown in Fig. 3.c. The proof of the theorem, the study of calibration dependency, and the complete algorithm for T-mAP evaluation are provided in Appendix B.

### 4.3 T-mAP Hyperparameters

T-mAP has two hyperparameters: the maximum allowed time delta $\delta$ and evaluation horizon $H$. We set $\delta$ twice the cost of the OTD because when the model predicts an incorrect label, OTD removes the prediction and adds the ground truth event with the total cost equal to $2C$. The horizon $H$ must be larger than $\delta$ to evaluate timestamp quality adequately. Therefore, depending on the dataset, we select $H$ to be approximately 3-7 times larger than $\delta$, ensuring the horizon captures an average of 6-15 events. The empirical study of T-mAP hyperparameters is presented in Figure 4. As shown, the chosen $\delta$ parameter is positioned to the right of the initial slope of the parameter-quality curve, ensuring an optimal balance between time granularity and task difficulty.

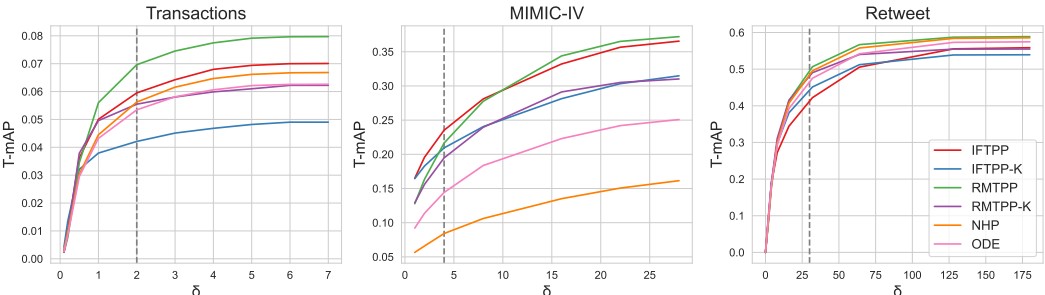

Figure 4: T-mAP dependency on the $\delta$ parameter. The dashed line indicates the selected value. Results for the StackOverflow and Amazon datasets are provided in the Appendix E.

## 5 HoTPP Benchmark

The HoTPP benchmark integrates data preprocessing, training, and evaluation in a single toolbox. Unlike previous benchmarks, HoTPP introduces the novel T-mAP metric for long-horizon prediction. HoTPP differs from prior MTPP benchmarks by including simple rule-based baselines and next-k models, simultaneously predicting multiple future events. The HoTPP benchmark is designed to focus on simplicity, extensibility, evaluation stability, reproducibility, and computational efficiency.

**Simplicity and Extensibility.** The benchmark code is organized into a clear structure, separating the core library, dataset-specific scripts, and configuration files. New methods can be integrated at various levels, including the configuration file, model architecture, loss function, metric, and training module. Core components are reusable through the Hydra configuration library (Yadan, 2019). For example, HoTPP can apply the HYPRO rescoring method to backbone models such as IFTPP, RMTPP, NHP, and ODE. Another example is our Next-K implementation, which can be applied to IFTTP or RMTPP.

**Evaluation Stability.** In the MTPP domain, many datasets contain only a few thousand sequences for training and evaluation. For example, the StackOverflow test set includes 401 sequences, Retweet contains 1956, and Amazon has 1851 sequences for testing. Previous long-horizon evaluation pipelines (Xue et al., 2022; 2023) made predictions only at the end of each sequence, resulting in a limited number of predictions and reduced evaluation stability. To address this limitation, we evaluate long-horizon predictions at multiple intermediate points.

**Reproducibility.** The HoTPP benchmark ensures reproducibility at multiple levels. First, we use the PytorchLightning library (Falcon & The PyTorch Lightning team, 2019) for training with a specified random seed and report multi-seed evaluation results. Second, data preprocessing is carefully designed to ensure datasets are constructed reproducibly. Finally, we specify the environment in a Dockerfile.

**Computational Efficiency.** Some methods are particularly slow, especially during autoregressive inference on large datasets. Straightforward evaluation can take several hours on a single Nvidia V100 GPU. We optimized the training and inference pipelines in two ways to accelerate computation. First, we implemented an efficient RNN that reuses computations during parallel autoregressive

Table 1: Evaluation results. The best result is shown in bold. The mean and standard deviation of each metric computed during five runs with different random seeds are reported.

| | Method | Mean length | Next-item | | | Long-horizon | |
|---|---|---|---|---|---|---|---|
| | | | Acc (%) | mAP (%) | MAE | OTD Val / Test | T-mAP Val / Test (%) |
| **Transactions** | MostPopular | 7.5 | $32.78_{\pm 0.00}$ | $0.86_{\pm 0.00}$ | $0.752_{\pm 0.000}$ | $7.37_{\pm 0.00}$ / $7.38_{\pm 0.00}$ | $1.06_{\pm 0.00}$ / $0.99_{\pm 0.00}$ |
| | Last 5 | 4.7 | $19.60_{\pm 0.00}$ | $0.87_{\pm 0.00}$ | $0.924_{\pm 0.000}$ | $7.40_{\pm 0.00}$ / $7.44_{\pm 0.00}$ | $2.46_{\pm 0.00}$ / $2.49_{\pm 0.00}$ |
| | IFTPP | 11.0 | $34.08_{\pm 0.04}$ | $\mathbf{3.47_{\pm 0.01}}$ | $\mathbf{0.693_{\pm 0.000}}$ | $6.88_{\pm 0.01}$ / $6.90_{\pm 0.01}$ | $5.82_{\pm 0.27}$ / $5.88_{\pm 0.13}$ |
| | IFTPP-K | 10.1 | $33.69_{\pm 0.03}$ | $3.25_{\pm 0.01}$ | $0.698_{\pm 0.001}$ | $7.18_{\pm 0.00}$ / $7.19_{\pm 0.00}$ | $4.42_{\pm 0.15}$ / $4.43_{\pm 0.16}$ |
| | RMTPP | 7.5 | $34.15_{\pm 0.07}$ | $\mathbf{3.47_{\pm 0.02}}$ | $0.749_{\pm 0.005}$ | $\mathbf{6.86_{\pm 0.01}}$ / $\mathbf{6.88_{\pm 0.01}}$ | $7.08_{\pm 0.16}$ / $6.69_{\pm 0.12}$ |
| | RMTPP-K | 7.3 | $33.63_{\pm 0.07}$ | $3.24_{\pm 0.02}$ | $0.749_{\pm 0.001}$ | $7.10_{\pm 0.01}$ / $7.11_{\pm 0.00}$ | $5.82_{\pm 0.05}$ / $5.52_{\pm 0.13}$ |
| | NHP | 9.4 | $35.43_{\pm 0.01}$ | $3.41_{\pm 0.01}$ | $0.696_{\pm 0.002}$ | $6.97_{\pm 0.01}$ / $6.98_{\pm 0.01}$ | $5.59_{\pm 0.07}$ / $5.61_{\pm 0.05}$ |
| | AttNHP | 7.6 | $31.12_{\pm x.xx}$ | $1.21_{\pm x.xx}$ | $0.717_{\pm x.xxx}$ | $7.52_{\pm x.xx}$ / $7.50_{\pm x.xx}$ | $1.71_{\pm x.xx}$ / $1.48_{\pm x.xx}$ |
| | ODE | 9.1 | $\mathbf{35.60_{\pm 0.06}}$ | $3.34_{\pm 0.06}$ | $0.695_{\pm 0.002}$ | $6.96_{\pm 0.01}$ / $6.97_{\pm 0.01}$ | $5.53_{\pm 0.08}$ / $5.52_{\pm 0.13}$ |
| | HYPRO | 6.9 | $34.26_{\pm x.xx}$ | $3.46_{\pm x.xx}$ | $0.758_{\pm x.xxx}$ | $7.04_{\pm x.xx}$ / $7.05_{\pm x.xx}$ | $\mathbf{7.79_{\pm x.xx}}$ / $\mathbf{7.05_{\pm x.xx}}$ |
| **MIMIC-IV** | MostPopular | 10.3 | $4.77_{\pm 0.00}$ | $2.75_{\pm 0.00}$ | $14.52_{\pm 0.00}$ | $19.82_{\pm 0.00}$ / $19.82_{\pm 0.00}$ | $0.55_{\pm 0.00}$ / $0.54_{\pm 0.00}$ |
| | Last 5 | 4.0 | $1.02_{\pm 0.00}$ | $2.63_{\pm 0.00}$ | $5.34_{\pm 0.00}$ | $19.75_{\pm 0.00}$ / $19.73_{\pm 0.00}$ | $2.41_{\pm 0.00}$ / $2.49_{\pm 0.00}$ |
| | IFTPP | 12.7 | $\mathbf{58.59_{\pm 0.02}}$ | $\mathbf{47.32_{\pm 1.39}}$ | $3.00_{\pm 0.01}$ | $\mathbf{11.51_{\pm 0.01}}$ / $\mathbf{11.53_{\pm 0.01}}$ | $21.93_{\pm 0.21}$ / $21.67_{\pm 0.21}$ |
| | IFTPP-K | 15.1 | $57.91_{\pm 0.13}$ | $44.60_{\pm 0.14}$ | $3.07_{\pm 0.02}$ | $13.17_{\pm 0.05}$ / $13.18_{\pm 0.05}$ | $22.46_{\pm 0.03}$ / $22.30_{\pm 0.03}$ |
| | RMTPP | 10.0 | $58.33_{\pm 0.02}$ | $46.24_{\pm 1.18}$ | $3.89_{\pm 0.03}$ | $13.64_{\pm 0.23}$ / $13.71_{\pm 0.03}$ | $21.49_{\pm 0.31}$ / $21.08_{\pm 0.29}$ |
| | RMTPP-K | 8.4 | $57.48_{\pm 0.06}$ | $43.47_{\pm 0.11}$ | $3.62_{\pm 0.02}$ | $14.68_{\pm 0.01}$ / $14.72_{\pm 0.02}$ | $20.70_{\pm 0.16}$ / $20.39_{\pm 0.14}$ |
| | NHP | 6.1 | $24.97_{\pm 0.94}$ | $11.12_{\pm 0.49}$ | $6.53_{\pm 0.77}$ | $18.59_{\pm 0.18}$ / $18.60_{\pm 0.19}$ | $7.26_{\pm 1.35}$ / $7.32_{\pm 1.33}$ |
| | AttNHP | 5.1 | $42.75_{\pm 0.52}$ | $28.55_{\pm 1.06}$ | $3.08_{\pm 0.04}$ | $14.66_{\pm 0.08}$ / $14.68_{\pm 0.08}$ | $\mathbf{22.64_{\pm 0.41}}$ / $\mathbf{22.46_{\pm 0.40}}$ |
| | ODE | 12.5 | $43.21_{\pm 2.30}$ | $25.34_{\pm 3.05}$ | $\mathbf{2.93_{\pm 0.03}}$ | $14.71_{\pm 0.34}$ / $14.74_{\pm 0.34}$ | $15.41_{\pm 0.21}$ / $15.18_{\pm 0.15}$ |
| | HYPRO | 7.4 | $58.35_{\pm x.xx}$ | $45.45_{\pm x.xx}$ | $3.95_{\pm x.xx}$ | $14.82_{\pm x.xx}$ / $14.87_{\pm x.xx}$ | $16.94_{\pm x.xx}$ / $16.77_{\pm x.xx}$ |
| **Retweet** | MostPopular | 28.0 | $58.50_{\pm 0.00}$ | $39.85_{\pm 0.00}$ | $18.82_{\pm 0.00}$ | $174.9_{\pm 0.0}$ / $173.5_{\pm 0.0}$ | $25.15_{\pm 0.00}$ / $23.91_{\pm 0.00}$ |
| | Last 10 | 9.8 | $50.29_{\pm 0.00}$ | $35.73_{\pm 0.00}$ | $21.87_{\pm 0.00}$ | $\mathbf{152.3_{\pm 0.0}}$ / $\mathbf{150.3_{\pm 0.0}}$ | $29.12_{\pm 0.00}$ / $29.24_{\pm 0.00}$ |
| | IFTPP | 26.1 | $59.95_{\pm 0.03}$ | $46.53_{\pm 0.04}$ | $18.27_{\pm 0.03}$ | $173.3_{\pm 4.1}$ / $172.7_{\pm 4.4}$ | $34.90_{\pm 4.63}$ / $31.75_{\pm 4.44}$ |
| | IFTPP-K | 14.8 | $59.55_{\pm 0.26}$ | $45.09_{\pm 0.62}$ | $\mathbf{18.21_{\pm 0.42}}$ | $168.6_{\pm 2.4}$ / $167.9_{\pm 2.8}$ | $37.11_{\pm 4.91}$ / $34.73_{\pm 5.11}$ |
| | RMTPP | 19.0 | $60.07_{\pm 0.10}$ | $46.81_{\pm 0.06}$ | $18.45_{\pm 0.16}$ | $167.6_{\pm 3.4}$ / $166.7_{\pm 3.3}$ | $47.86_{\pm 0.91}$ / $44.74_{\pm 0.94}$ |
| | RMTPP-K | 13.7 | $59.99_{\pm 0.05}$ | $46.34_{\pm 0.09}$ | $18.33_{\pm 0.13}$ | $164.7_{\pm 2.8}$ / $163.9_{\pm 2.8}$ | $49.07_{\pm 1.16}$ / $46.16_{\pm 1.32}$ |
| | NHP | 18.1 | $\mathbf{60.08_{\pm 0.04}}$ | $\mathbf{46.83_{\pm 0.11}}$ | $18.42_{\pm 0.11}$ | $167.0_{\pm 1.6}$ / $165.8_{\pm 1.6}$ | $48.31_{\pm 0.67}$ / $45.07_{\pm 0.34}$ |
| | AttNHP | 29.5 | $60.03_{\pm 0.03}$ | $46.74_{\pm 0.01}$ | $18.39_{\pm 0.05}$ | $173.3_{\pm 1.0}$ / $171.6_{\pm 1.0}$ | $28.32_{\pm 1.24}$ / $25.85_{\pm 1.08}$ |
| | ODE | 19.0 | $59.95_{\pm 0.08}$ | $46.65_{\pm 0.04}$ | $18.38_{\pm 0.04}$ | $166.5_{\pm 0.5}$ / $165.3_{\pm 0.5}$ | $48.70_{\pm 0.93}$ / $44.81_{\pm 0.69}$ |
| | HYPRO | 15.0 | $59.87_{\pm x.xx}$ | $46.69_{\pm x.xx}$ | $18.75_{\pm x.xx}$ | $171.4_{\pm x.x}$ / $170.7_{\pm x.x}$ | $\mathbf{49.90_{\pm x.xx}}$ / $\mathbf{46.99_{\pm x.xx}}$ |
| **Amazon** | MostPopular | 20.6 | $33.46_{\pm 0.00}$ | $9.58_{\pm 0.00}$ | $0.304_{\pm 0.000}$ | $7.20_{\pm 0.00}$ / $7.18_{\pm 0.00}$ | $8.59_{\pm 0.00}$ / $8.31_{\pm 0.00}$ |
| | Last 5 | 5.0 | $24.23_{\pm 0.00}$ | $8.15_{\pm 0.00}$ | $0.321_{\pm 0.000}$ | $\mathbf{6.43_{\pm 0.00}}$ / $\mathbf{6.41_{\pm 0.00}}$ | $9.73_{\pm 0.00}$ / $9.21_{\pm 0.00}$ |
| | IFTPP | 13.7 | $35.73_{\pm 0.12}$ | $17.14_{\pm 0.10}$ | $\mathbf{0.242_{\pm 0.003}}$ | $6.58_{\pm 0.10}$ / $6.52_{\pm 0.05}$ | $21.94_{\pm 0.46}$ / $22.56_{\pm 0.52}$ |
| | IFTPP-K | 13.3 | $35.11_{\pm 0.08}$ | $16.48_{\pm 0.03}$ | $0.246_{\pm 0.000}$ | $6.72_{\pm 0.01}$ / $6.68_{\pm 0.01}$ | $22.06_{\pm 0.05}$ / $22.57_{\pm 0.07}$ |
| | RMTPP | 16.7 | $\mathbf{35.76_{\pm 0.06}}$ | $\mathbf{17.21_{\pm 0.02}}$ | $0.294_{\pm 0.001}$ | $6.62_{\pm 0.02}$ / $6.57_{\pm 0.03}$ | $19.70_{\pm 0.31}$ / $20.06_{\pm 0.33}$ |
| | RMTPP-K | 15.8 | $35.06_{\pm 0.14}$ | $16.37_{\pm 0.20}$ | $0.300_{\pm 0.005}$ | $6.92_{\pm 0.01}$ / $6.87_{\pm 0.02}$ | $17.85_{\pm 0.29}$ / $18.12_{\pm 0.30}$ |
| | NHP | 9.9 | $11.06_{\pm 3.43}$ | $11.22_{\pm 0.19}$ | $0.449_{\pm 0.014}$ | $9.04_{\pm 0.31}$ / $9.02_{\pm 0.35}$ | $\mathbf{26.24_{\pm 0.36}}$ / $\mathbf{26.29_{\pm 0.55}}$ |
| | AttNHP | 18.5 | $31.83_{\pm 0.32}$ | $9.70_{\pm 0.28}$ | $0.461_{\pm 0.003}$ | $7.32_{\pm 0.06}$ / $7.30_{\pm 0.06}$ | $14.50_{\pm 0.58}$ / $14.62_{\pm 0.80}$ |
| | ODE | 9.6 | $7.54_{\pm 0.95}$ | $10.14_{\pm 0.23}$ | $0.492_{\pm 0.018}$ | $9.48_{\pm 0.11}$ / $9.46_{\pm 0.08}$ | $23.54_{\pm 0.62}$ / $22.96_{\pm 0.61}$ |
| | HYPRO | 18.0 | $35.69_{\pm x.xx}$ | $\mathbf{17.21_{\pm x.xx}}$ | $0.295_{\pm x.xxx}$ | $6.63_{\pm x.xx}$ / $6.61_{\pm x.xx}$ | $20.58_{\pm x.xx}$ / $20.53_{\pm x.xx}$ |
| **StackOverflow** | MostPopular | 14.0 | $42.90_{\pm 0.00}$ | $5.45_{\pm 0.00}$ | $0.744_{\pm 0.000}$ | $13.56_{\pm 0.00}$ / $13.77_{\pm 0.00}$ | $6.10_{\pm 0.00}$ / $5.56_{\pm 0.00}$ |
| | Last 10 | 9.3 | $26.42_{\pm 0.00}$ | $5.20_{\pm 0.00}$ | $0.934_{\pm 0.000}$ | $14.52_{\pm 0.00}$ / $14.55_{\pm 0.00}$ | $8.67_{\pm 0.00}$ / $6.72_{\pm 0.00}$ |
| | IFTPP | 24.1 | $45.41_{\pm 0.11}$ | $13.00_{\pm 0.79}$ | $\mathbf{0.641_{\pm 0.002}}$ | $13.57_{\pm 0.07}$ / $13.64_{\pm 0.05}$ | $8.78_{\pm 0.87}$ / $8.31_{\pm 0.50}$ |
| | IFTPP-K | 15.3 | $44.85_{\pm 0.24}$ | $11.16_{\pm 1.07}$ | $0.644_{\pm 0.003}$ | $13.41_{\pm 0.07}$ / $13.51_{\pm 0.06}$ | $12.42_{\pm 0.97}$ / $11.42_{\pm 0.78}$ |
| | RMTPP | 14.5 | $\mathbf{45.43_{\pm 0.13}}$ | $\mathbf{13.33_{\pm 0.25}}$ | $0.701_{\pm 0.007}$ | $12.95_{\pm 0.02}$ / $13.17_{\pm 0.05}$ | $13.26_{\pm 0.29}$ / $12.72_{\pm 0.16}$ |
| | RMTPP-K | 12.0 | $44.89_{\pm 0.09}$ | $11.72_{\pm 0.10}$ | $0.689_{\pm 0.001}$ | $\mathbf{12.92_{\pm 0.01}}$ / $\mathbf{13.13_{\pm 0.01}}$ | $14.91_{\pm 0.19}$ / $14.30_{\pm 0.09}$ |
| | NHP | 13.0 | $44.53_{\pm 0.05}$ | $10.86_{\pm 0.19}$ | $0.715_{\pm 0.004}$ | $13.02_{\pm 0.02}$ / $13.24_{\pm 0.02}$ | $12.67_{\pm 0.55}$ / $11.96_{\pm 0.40}$ |
| | AttNHP | 15.5 | $45.17_{\pm 0.10}$ | $12.67_{\pm 0.13}$ | $0.705_{\pm 0.001}$ | $13.08_{\pm 0.03}$ / $13.30_{\pm 0.02}$ | $11.95_{\pm 0.19}$ / $11.13_{\pm 0.32}$ |
| | ODE | 13.9 | $44.38_{\pm 0.09}$ | $10.12_{\pm 0.19}$ | $0.711_{\pm 0.004}$ | $13.04_{\pm 0.02}$ / $13.27_{\pm 0.03}$ | $11.37_{\pm 0.32}$ / $10.52_{\pm 0.23}$ |
| | HYPRO | 11.8 | $45.18_{\pm x.xx}$ | $12.88_{\pm x.xx}$ | $0.715_{\pm x.xxx}$ | $13.04_{\pm x.xx}$ / $13.26_{\pm x.xx}$ | $\mathbf{15.57_{\pm x.xx}}$ / $\mathbf{14.69_{\pm x.xx}}$ |

inference from multiple starting positions. Second, we developed highly optimized versions of the thinning algorithm used for sampling in NHP and continuous-time neural architectures (NHP and ODE), achieving up to a 4x performance improvement compared to the official implementations. These optimizations allowed us to conduct the first large-scale evaluations of algorithms such as NHP, AttNHP, ODE, and HYPRO on datasets like Transactions and MIMIC-IV. Additionally, we provide the first open-source CUDA implementation of the batched linear assignment solver, significantly enhancing the applicability of the proposed T-mAP metric.

A detailed description of the benchmark can be found in Appendix C, with further details on HoTPP's computational performance in Appendix C.5.

# 6 EXPERIMENTS

We conduct experiments on five datasets of varying sizes and origins: the Transactions dataset (AI-Academy, 2021), the MIMIC-IV medical dataset (Johnson et al., 2023), two social networks datasets, Retweet (Zhao et al., 2015) and StackOverflow (Jure, 2014), and the Amazon reviews dataset (Jianmo, 2018). Dataset statistics and evaluation parameters are provided in Appendix C.4.

We evaluate representative modeling methods from different groups:

1. Rule-based baselines. **MostPopular** generates a constant output with the most popular label from the prefix and the average time step. The **Last N** baseline outputs N previous events with adjusted timestamps.

2. Intensity-free approaches. We implement the **IFTPP** method, which combines mean absolute error (MAE) of the time step prediction with cross-entropy categorical loss for labels (Shchur et al., 2019; Padhi et al., 2021; McDermott et al., 2024).

3. Intensity-based approaches. We implement **RMTPP** (Du et al., 2016) as an example of the TPP approach with a traditional RNN. We add **NHP** (Mei & Eisner, 2017), based on a continuous time LSTM architecture. We also evaluate the **AttNHP** approach that utilizes a continuous time transformer model (Yang et al., 2022) and **ODE** (Rubanova et al., 2019).

4. Next-K approaches. We adapt IFTPP and RMTPP to predict multiple future events directly, implementing **IFTPP-K** and **RMTPP-K**, respectively. These approaches originate in time series analysis and have not been previously applied in the MTPP domain.

5. Reranking. We implement **HYPRO** (Xue et al., 2022), which generates multiple hypotheses and selects the best sequence using a contrastive approach.

Additional details on these methods are provided in Appendix A, with training specifics outlined in Appendix D. Hyperparameters for both the methods and metrics are presented in Appendix E. The main evaluation results are shown in Table 1. In the following sections, we will discuss these results from various perspectives and offer additional analysis of the methods' behavior.

## 6.1 METRICS COMPARISON

The evaluation results in Table 1 show that the OTD metric can yield low values even for simple rule-based baselines. For instance, the Last baseline achieves the lowest OTD values on the Retweet and Amazon datasets. In contrast, the T-mAP metric more clearly distinguishes between rule-based baselines and deep learning models. This difference arises from the ability to measure model confidence. OTD evaluates predicted labels alone, while T-mAP measures average precision by adjusting logits thresholds. Unlike rule-based baselines, deep models provide confidence scores, whereas baselines output only labels, resulting in low T-mAP scores for simple baselines.

The HYPRO and NHP methods maximize T-mAP, while statistical baselines, IFTPP, and RMTPP achieve the lowest OTD values. The difference between the two metrics lies in balancing the importance of event order versus time prediction quality. OTD evaluates the first 5-10 events, and to minimize OTD, a model must correctly identify the events that constitute the beginning of the sequence. IFTPP and RMTPP jointly predict the next event's time and label, leading to better ordering. In contrast, methods like NHP, AttNHP, and ODE predict time steps independently for each event type. When the differences between timestamps are small, these methods can result in random ordering, negatively affecting the OTD score. T-mAP, on the other hand, is less sensitive to event order as long as the time predictions are accurate.

The HYPRO method, designed to differentiate between ground truth and predicted sequences using a discriminator model, achieves high T-mAP scores. This highlights T-mAP's ability to reward models that generate realistic sequences. We also observed that methods with high T-mAP scores often have low next-event prediction accuracy. For instance, on the Retweet, Amazon, and MIMIC-IV datasets, high long-horizon performance is usually accompanied by low next-event prediction scores. This is particularly true for the HYPRO method, whose loss function is heavily focused on long-horizon quality. The contrast on the MIMIC-IV dataset can be attributed to the large number of events with identical timestamps, making precise event ordering more difficult and impacting both next-event and OTD scores.

## 6.2 INTENSITY-BASED VS INTENSITY-FREE

Previous works mostly compared either intensity-based neural methods (Du et al., 2016; Xue et al., 2022) or intensity-free approaches (Padhi et al., 2021; McDermott et al., 2024). The only exception is IFTPP (Shchur et al., 2019), which did not compare with NHP and included only test set likelihood as an evaluation metric. This raises the question: which approach is superior? We compare the intensity-free IFTPP method with intensity-based RMTPP, NHP, and ODE approaches. Table 1 shows that the intensity-free IFTPP method excels in the next-event MAE prediction, as it optimizes this metric. In other scenarios, the results vary between datasets. Therefore, we conclude that there is no preferred solution, and both approaches warrant attention in future research.

## 6.3 AUTOREGRESSION VS DIRECT HORIZON PREDICTION

We compared the IFTPP and RMTPP methods with their Next-K counterparts. Our results indicate that autoregressive models generally perform better regarding next-event prediction quality, while Next-K approaches improve T-mAP scores on all datasets except Transactions. The OTD-based rankings vary across datasets, with no single approach consistently outperforming the others. Despite these findings, little effort has been made to adapt Next-K models to MTPPs, suggesting future research in this area.

We also observed that the Next-K approaches exhibit some interesting properties compared to autoregression. For instance, the entropy of label distributions in autoregression models decreases with increasing generation steps, as shown in Figure 5. The likely reason for this behavior is the dependency of the model's future output on its past errors. Conversely, Next-K models demonstrate increasing entropy, which is expected as future events become harder to predict. This suggests that Next-K models have the potential to predict better the distributions of future labels, a factor that should be considered in future research. Appendix G provides a qualitative analysis of predictions diversity.

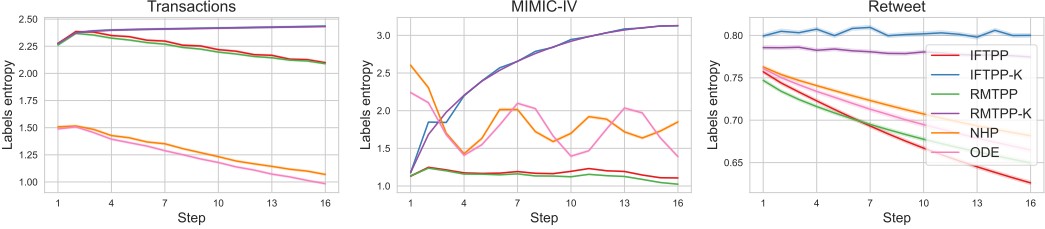

Figure 5: Entropy of label distributions as a function of the position in the generated sequence. Results for StackOverflow and Amazon datasets are provided in Appendix J.

## 6.4 THE MAXIMUM SEQUENCE LENGTH

We observed that long-horizon prediction quality largely depends on the maximum number of allowed predictions in autoregression and Next-K models. As shown in Fig. 6, the optimal number of predicted events is usually less than the maximum number of events on the horizon. For example, in the Transactions dataset, the optimal length ranges from 3 to 5, depending on the method. This indicates that the quality of confidence estimation degrades for events further in the future, therefore it is beneficial to limit the number of predictions manually.

Therefore, we conclude that greater attention should be given to probability calibration between generation steps. This finding also highlights progress in long-horizon prediction tasks, particularly in determining the maximum horizon that can be accurately modeled. T-mAP is the only metric capable of evaluating the optimal predicted sequence length, while the OTD metric remains unaffected by sequence length as it compares fixed-size prefixes.

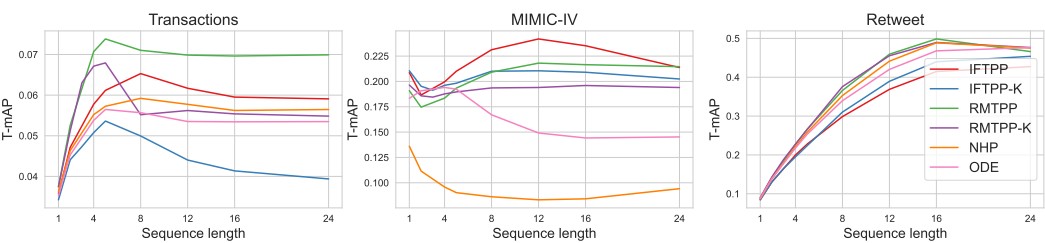

Figure 6: T-mAP dependency on the maximum length of the predicted sequence. Results for the StackOverflow and Amazon datasets are detailed in Appendix K.

## 7 LIMITATIONS AND FUTURE WORK

The proposed benchmark has several shortcomings. The training process for some methods is relatively slow. For example, a continuous time LSTM from the NHP method lacks an effective open-source GPU implementation, which could be developed in the future. Autoregression can also be optimized using specialized GPU implementations, which would significantly impact HYPRO training. Additionally, the list of implemented backbone architectures and losses can be extended, for example, by incorporating hybrid (Deshpande et al., 2021) and diffusion models (Zhou et al., 2023).

Our benchmark encourages future research to develop improved techniques for predicting future events and establish simple baselines for better measurement of progress in the field. For example, Next-K models, which predict multiple future events without autoregression, show promising results in long-horizon prediction tasks. We suggest further exploration of these models. We also highlight the importance of label distribution estimation and emphasize the need to improve confidence estimation and calibration.

The T-mAP metric can be applied in domains beyond event sequences, like action recognition (Su & Hua, 2017). T-mAP can potentially offer better timestamp evaluation in these domains than evaluating indices with OPW. T-mAP can also provide a more natural hyperparameter selection regarding the modeling horizon and maximum allowed time error. Our theoretical justification of T-mAP can potentially be adapted to the mAP estimation algorithms in computer vision.

## 8 CONCLUSION

In this paper, we propose the HoTPP benchmark to assess the quality of long-horizon events forecasting. Our extensive evaluations using established datasets and various predictive models reveal a critical insight: high accuracy in next-event prediction does not necessarily correlate with superior performance in horizon prediction. This finding underscores the need for benchmarks like HoTPP, emphasizing the importance of long-horizon accuracy and robustness. By shifting the focus from short-term to long-term predictive capabilities, HoTPP aims to drive the development of more sophisticated and reliable event sequence prediction models. This, in turn, has the potential to significantly enhance the practical applications of sequential event prediction in various domains, fostering innovation and improving decision-making processes across a wide range of industries.

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

## A    BACKGROUND ON MARKED TEMPORAL POINT PROCESSES MODELING

**Intensity-based approaches.** Modeling the probability density function (PDF) $f^*(t_i) = f(t_i|t_1, \ldots, t_{i-1})$ for the next event time is typically challenging, as it requires the additional constraint that its integral equals one. Instead, the non-negative intensity function $\lambda(t_i) \geq 0$ is usually modeled. The following equation gives the relationship between the PDF and the intensity function:

$$f^*(t_i) = \lambda(t_i) \exp\left(-\int_{t_{i-1}}^{t_i} \lambda(s)ds\right). \tag{4}$$

The derivation is provided in (Rizoiu et al., 2017). Different TPPs are characterized by their intensity function $\lambda(t_i)$. In Poisson and non-homogeneous Poisson processes, the intensity function is independent of previous events, meaning event occurrences depend solely on external factors. In contrast, self-exciting processes are characterized by previous events increasing the intensity of future events. A notable example of a self-exciting process is the Hawkes process, in which each event linearly affects the future intensity:

$$\lambda(t_i) = \lambda_0(t_i) + \sum_{k=1}^{i-1} \phi(t_i - t_k), \tag{5}$$

where $\lambda_0(t_i) \geq 0$ is the base intensity function, independent of previous events, and $\phi(x) \geq 0$ is the so-called *memory kernel* function. Given the intensity function $\lambda(t_i)$, predictions are typically made by sampling or expectation estimation. Sampling is usually performed using the *thinning algorithm*, a specific rejection sampling approach. For details on the implementation of sampling, please refer to (Rizoiu et al., 2017).

Recent research has focused on modeling complex intensity functions. Various neural network architectures have been adapted to address this problem. These approaches differ in the type of neural network used and the model of the intensity function. Neural architectures range from simple RNNs and Transformers to specially designed continuous-time models like NHP (Mei & Eisner, 2017) and Neural ODEs (Rubanova et al., 2019). The intensity function between events can be modeled as a sum of intensities from previous events, as in Hawkes processes, or by directly predicting the inter-event intensity given the context embedding.

**Intensity-free modeling.** Some methods evaluate the next event time distribution without using intensity functions. For example, intensity-free (Shchur et al., 2019) represents the distribution as a mixture of Gaussians or through normalizing flows. Instead of predicting the distribution directly, other approaches solve a regression problem using MAE or RMSE loss. However, it can be shown that both MAE and RMSE losses are closely related to distribution prediction. Specifically, RMSE is analogous to log-likelihood optimization with a Normal distribution, and MAE is similar to the log-likelihood of a Laplace distribution (Bishop, 1994). In our experiments, we evaluated a model trained with MAE loss as an example of an intensity-free method.

**Rescoring with HYPRO.** HYPRO (Xue et al., 2022) is an extension applicable to any sequence prediction method capable of sampling. HYPRO takes a pretrained generative model and trains an additional scoring module to select the best sequence from a sample. It is trained with a contrastive loss to distinguish between the generated sequence and the ground truth. HYPRO generates multiple sequences with a background model during inference and selects the best one by maximizing the estimated score. Although HYPRO is intended to improve quality compared to simple sampling, it is unclear whether HYPRO outperforms expectation-based prediction. In our work, we apply HYPRO to the outputs of the RMTPP model.

**Next-K models.** While not commonly used in the MTPP field, Next-K approaches are popular in time series modeling (Lim & Zohren, 2021). These methods predict multiple future events at once, avoiding the need for autoregressive inference. The advantages of Next-K approaches include fast inference and stability, as predictions do not depend on potential errors from previous steps, unlike autoregression. The main limitation is a fixed prediction horizon, as the model cannot predict sequences of arbitrary length. However, applying a modified autoregressive approach can potentially address this limitation.

In our work, we implement a Next-K variant of the IFTPP model, which is straightforward. Given K predictions and K ground truth events, we compute MAE and cross-entropy losses between corresponding pairs of events from both sequences. The Next-K variant of the RMTPP approach is more complex, as it violates Hawkes assumptions: the $i$-th prediction doesn't depend on predictions $1 \ldots i-1$. Consequently, RMTPP-K, unlike RMTPP, cannot model dependencies between the predicted K events. Despite this, RMTPP-K still performs strongly, particularly on the StackOverflow dataset.

## B   T-mAP Computation and Proofs

**Definitions and scope.** In this section, we consider weighted bipartite graphs. The first part consists of *predictions*, and the second consists of *ground truth* events. We assume that all edges connected to the same prediction have the same weight. A matching is a set of edges $(a_i, b_i)$ between predictions and ground truth events. A matching is termed *optimal* if it (1) has the maximum size among all possible matching and (2) has the minimum total weight among all matches of that size. We denote all optimal matchings in the graph $\mathcal{G}$ as $M(\mathcal{G})$.

**Lemma A.** *Consider a graph $\mathcal{G}'$, constructed from a graph $\mathcal{G}$ by adding one ground truth vertex with corresponding weighted edges. Then any optimal matching $m' \in M(\mathcal{G}')$ will either have a size greater than the sizes of matchings in $M(\mathcal{G})$ or have a total weight equal to that of matchings in $M(\mathcal{G})$.*

$S_p$:

$S_{gt}$:

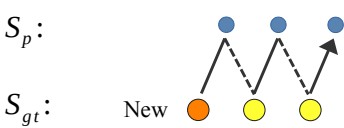 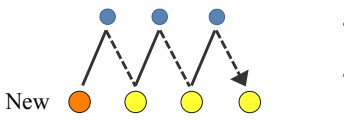

New                          New

$---\cdot\ m \in M(\mathcal{G})$

$----\ m' \in M(\mathcal{G}')$

a. The process stops on a prediction          b. The process stops on a ground truth

Figure 7: Illustration of the iterative process in Lemma A.

*Proof.* If the size of $m'$ is greater than the size of matchings in $M(\mathcal{G})$, then the lemma holds. The optimal matching in the extended graph $\mathcal{G}'$ cannot have a size smaller than the optimal matching in $\mathcal{G}$. Now, consider the case when both matchings have the same size. We will show that $m'$ has a total weight equal to that of the matchings in $M(\mathcal{G})$.

Denote $b$ the new ground truth event in graph $\mathcal{G}'$. If $m'$ does not include vertex $b$, then $m'$ is also optimal in $\mathcal{G}$, and the lemma holds. If $m'$ includes vertex $b$, then we can walk through the graph with the following process:

1. Start from the new vertex $b$.

2. If we are at a ground truth event $b_i$ and there is an edge $(a_i, b_i)$ in matching $m'$, move to the vertex $a_i$.

3. If we are at a prediction $a_i$ and there is an edge $(a_i, b_i)$ in matching $m$, move to the vertex $b_i$.

Example processes are illustrated in Fig. 7. The vertices in each process do not repeat; otherwise, there would be two edges in either $m$ or $m'$ with the same vertex, which contradicts the definition of a matching.

If the process finishes at a predicted event, then $m'$ has a size greater than the size of $m$, and the lemma holds. If the process finishes at a ground truth event, we can replace a part of matching $m'$ traced with a part of matching $m$. The resulting matching $\hat{m}$ will have a size equal to $m$ and $m'$ and a total weight equal to both matchings. This proves the lemma. □

**Theorem 4.1.** *For any threshold $h$, there exists an optimal matching in the graph $\mathcal{G}_h$ such that it is a subset of an optimal matching in the full graph $\mathcal{G}$:*

$$\forall h \forall m \in M(\mathcal{G}) \exists m_h \in M(\mathcal{G}_h) : m_h \subset m. \tag{6}$$

*Proof.* If the threshold is lower than any score $s_i^p$, then $\mathcal{G}_h = \mathcal{G}$, and $m_h = m$ satisfies the theorem. Otherwise, some predictions are filtered by the threshold and $\mathcal{G}_h \subset \mathcal{G}$.

Without the loss of generality, assume that the threshold is low enough to filter out only one prediction. Otherwise, we can construct a chain of thresholds $h_1 < h_2 < \cdots < h$, with each subsequent threshold filtering an additional vertex, and apply the theorem iteratively.

Denote $i$ to be the only vertex present in $\mathcal{G}$ and filtered from $\mathcal{G}_h$. If there is no edge containing vertex $i$ in the matching $m(\mathcal{G})$, then $m(\mathcal{G})$ is also the optimal matching in $\mathcal{G}_h$ and theorem holds.

Consider the case when matching $m(\mathcal{G})$ contains an edge $(i, j)$. Let $\hat{m}_h = m(\mathcal{G}) \setminus \{(i, j)\}$, i.e., the matching without edge $(i, j)$. If it is optimal for the graph $\mathcal{G}_h$, then it will satisfy the theorem. Otherwise, there is an optimal matching $m_h$ with either (a) more vertices or (b) a smaller total weight than $\hat{m}_h$.

In (a), matching $m_h$ has a size greater than the size of $\hat{m}_h$. It follows that $m_h$ has the maximum size in the complete graph $\mathcal{G}$. If the total weight of $m_h$ is larger or equal to the optimal weight in $\mathcal{G}$, then it can not be less than the total weight of $\hat{m}_h$, which contradicts our assumption. At the same time, the total weight of $m_h$ can not be less than the optimal weight in the complete graph $\mathcal{G}$. It follows that case (a) leads to a contradiction.

Consider the case (b) when the size of $m_h$ equals $\hat{m}_h$. If $m_h$ does not include vertex $j$, then both $m_h$ and $\hat{m}_h$ are optimal. Otherwise, we can remove vertex $j$ from $\mathcal{G}_h$ and construct a new optimal matching $m'_h$ without $j$ by using Lemma A, and again, both matchings $m'_h$ and $\hat{m}_h$ are optimal. This concludes the proof. $\square$

**Algorithm.** Using the theorem, we can introduce an effective algorithm for T-mAP computation. T-mAP is defined on a batch of predictions $\{\mathcal{S}_p^i\}_{i=1}^n$ and ground truth sequences $\{\mathcal{S}_{gt}^i\}_{i=1}^n$. Let $\mathcal{S}_{gt}^{i,l}$ denote a subsequence of the ground truth sequence $\mathcal{S}_{gt}^i$ containing all events with label $l$. By definition, multiclass T-mAP is the average of the average precision (AP) values for each label $l$:

$$\text{T-mAP}(\{\mathcal{S}_p^i\}, \{\mathcal{S}_{gt}^i\}) = \frac{1}{L} \sum_{l=1}^{L} \text{AP}(\{\mathcal{S}_p^i\}, \{\mathcal{S}_{gt}^{i,l}\}). \tag{7}$$

Consider AP computation for a particular label $l$. AP is computed as the area under the precision-recall curve:

$$\text{AP} = \sum_i (\text{Rec}_i - \text{Rec}_{i-1}) \text{Prec}_i, \tag{8}$$

where $\text{Rec}_i$ is the $i$-th recall value in a sorted sequence and $\text{Prec}$ is the corresponding precision. Iteration is done among all distinct recall values ($\text{Rec}_0 = 0$).

We have several correct and incorrect predictions for each threshold $h$ on the predicted label probability. These quantities define precision, recall, and the total number of ground truth events. A prediction is correct if it has an assigned ground truth event within the required time interval $|t_i^p - t_j^{gt}| \leq \delta$. Note that each target can be assigned to at most one prediction. Therefore, we define a matching, i.e., the correspondence between predictions and ground truth events. The theorem states that the maximum size matching for each threshold $h$ can be constructed as a subset of an optimal matching between full sequences $\{\mathcal{S}_p^i\}_{i=1}^n$ and $\mathcal{S}_{gt}^{i,l}$, where optimal matching is a solution of the assignment problem with a bipartite graph, defined in Section 4.2.

As the matching, without loss of generality, can be considered constant for different thresholds, we can split all predictions into two parts: those that were assigned a ground truth and those that were unmatched. The matched set constitutes potential true or false positives, depending on the threshold. The unmatched set is always considered a false positive. Similarly, unmatched ground truth events are always considered false negatives. The resulting algorithm for AP computation for each label $l$ involves the following steps:

1. Compute the optimal matching between predictions and ground truth events.

2. Collect (a) scores of matched predictions, (b) scores of unmatched predictions, and (c) the number of unmatched ground truth events.

3. Assign a positive label to matched predictions and a negative label to unmatched ones.

4. Evaluate maximum recall as the fraction of matched ground truth events.

5. Find the area under the precision-recall curve for the constructed binary classification problem from item 3 and multiply it by the maximum recall value.

We have thus defined all necessary steps for T-mAP evaluation. Its complexity is $\mathcal{O}(LBN^3)$, where $L$ is the number of classes, $B$ is the number of sequence pairs, and $N$ is the maximum length of predicted and ground truth sequences.

**Calibration dependency.** Calibration influences the weights assigned to the edges of the graph $\mathcal{G}$. T-mAP computation involves two key steps for each class label: matching and AP estimation. While average precision (AP) is invariant to monotonic transformations of predicted class logits, the matching step is only invariant to linear transformations. Specifically, the optimal matching seeks to minimize the total weight in the following form:

$$C = \underset{m \in M(\mathcal{G})}{\arg\min} \sum_{(i,j) \in m} (-s_i^p). \tag{9}$$

A linear transformation of logits with a positive scaling factor will adjust the total weight accordingly, but the optimal matching will remain unchanged. Since the matching is performed independently for each class, we conclude that T-mAP is invariant to linear calibration.

## C  HoTPP Benchmark Details

### C.1  Architecture

HoTPP incorporates best practices from extensible and reproducible ML pipelines. It leverages PyTorch Lightning (Falcon & The PyTorch Lightning team, 2019) as the core training library, ensuring reproducibility and portability across various computing environments. Additionally, HoTPP utilizes the Hydra configuration library (Yadan, 2019) to enhance extensibility. The overall architecture is illustrated in Figure 8. HoTPP supports both discrete-time and continuous-time models as well as RNN and Transformer architectures. Implementing a new method requires only adding essential components, while the rest can be configured through Hydra and configuration files.

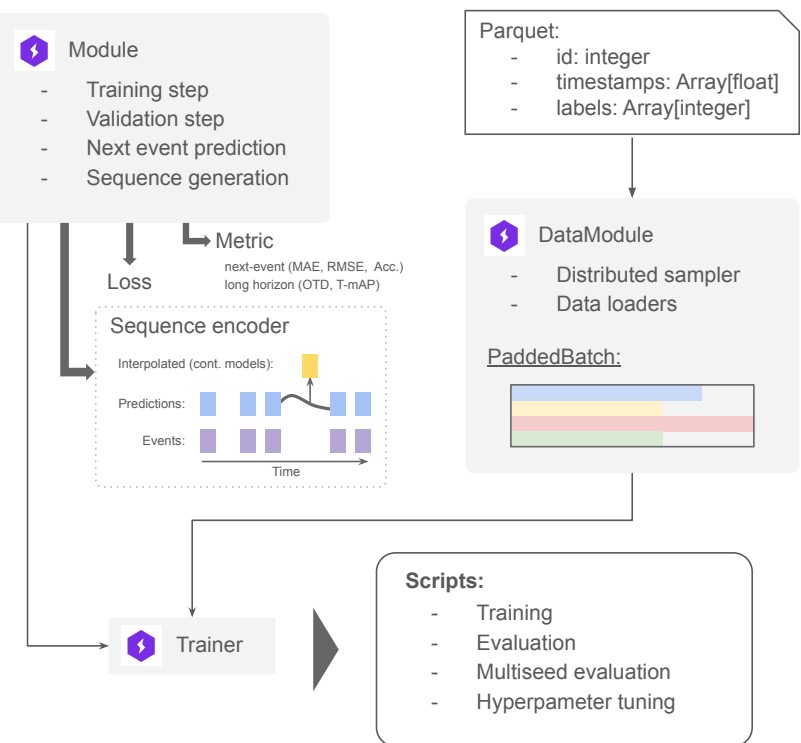

Figure 8: The architecture of the HoTPP library.

## C.2 METRICS

MTPP models are typically evaluated based on the accuracy of next-event predictions. Common metrics include mean absolute error (MAE) or root mean square error (RMSE) for time shifts and accuracy or error rate for label predictions. Some studies also assess test set likelihood as predicted by the model, but we do not include this measure as it is intractable for the IFTPP and Next-K models. Previous works have advanced long-horizon evaluation using OTD (Mei et al., 2019; Xue et al., 2023). In addition to these metrics, we evaluate the novel T-mAP metric, which addresses the shortcomings of previous metrics, as discussed in Section 3.

## C.3 BACKBONES

We use three types of architectures in our experiments. The first is the GRU network (Cho et al., 2014), one of the top-performing neural architectures for event sequences (Babaev et al., 2022). We also implement the continuous-time LSTM (CT-LSTM) from the NHP method (Mei & Eisner, 2017). Since CT-LSTM requires a specialized loss function and increases training time, we use it exclusively with the NHP method, preferring GRU in other cases. An additional advantage of GRU is that its output is equal to its hidden state, simplifying the estimation of intermediate hidden states for autoregressive inference starting from the middle of a sequence. Lastly, we implement the AttNHP continuous-time transformer model (Yang et al., 2022).

## C.4 DATASETS

For the first time, we combine domains such as financial transactions, social networks, and medical records into a single evaluation benchmark. Specifically, we provide evaluation results on a transactional dataset (Babaev et al., 2022), the MIMIC-IV medical dataset (Johnson et al., 2023), and social network datasets (Retweet (Zhao et al., 2015), StackOverflow (Jure, 2014), and Amazon (Jianmo, 2018)). These datasets represent diverse underlying processes: social network data is influenced by cascades (Zhao et al., 2015), medical records exhibit repetitive patterns and transactional data reflects daily activities, combining regularity with significant uncertainty.

The dataset statistics are presented in Table 2. The Transactions dataset has the longest average sequence length and the largest number of classes, while MIMIC-IV contains the highest number of sequences. Retweet is medium size, whereas Amazon and StackOverflow can be considered small datasets.

Table 2: Datasets statistics

| Dataset | Sequences | Events | Mean Length | Mean Horizon Length | Mean Duration | Classes |
|---|---|---|---|---|---|---|
| Transactions | 50k | 43.7M | 875 | 9.0 | 719 | 203 |
| MIMIC-IV | 120k | 2.4M | 19.7 | 6.6 | 503 | 34 |
| Retweet | 23k | 1.3M | 56.4 | 14.7 | 1805 | 3 |
| Amazon | 9k | 403K | 43.6 | 14.8 | 22.1 | 16 |
| StackOverflow | 2k | 138K | 64.2 | 12.0 | 55.3 | 22 |

Transactions[3], Retweet[4], Amazon[5], and StackOverflow[6] datasets were obtained from the Hugging-Face repository. Transactions data was released in competition and came with a free license[7]. Retweet, Amazon, and StackOverflow come with an Apache 2.0 license. MIMIC-IV is subject to PhysioNet Credentialed Health Data License 1.5.0, which requires ethical use of this dataset. Because of a complex data structure, we implement a custom preprocessing pipeline for the MIMIC-IV dataset.

---

[3] https://huggingface.co/datasets/dllllb/age-group-prediction
[4] https://huggingface.co/datasets/easytpp/retweet
[5] https://huggingface.co/datasets/easytpp/amazon
[6] https://huggingface.co/datasets/easytpp/stackoverflow
[7] https://www.kaggle.com/competitions/clients-age-group/data

**Notes on MIMIC-IV preprocessing**

MIMIC-IV is a publicly available electronic health record database that includes patient diagnoses, lab measurements, procedures, and treatments. We leverage the data preprocessing pipeline from EventStreamGPT to construct intermediate representations in EFGPT format. This process yields three key entities: a subjects data frame containing time-independent patient records, an events data frame listing event types occurring to subjects at specific timestamps, and a measurements data frame with time-dependent measurement values linked to the events data frame.

We combine events and measurements to create sequences of events for each subject. The classification labels are generally divided into two categories: diagnoses, represented by ICD codes, and event types, such as admissions, procedures, and measurements. A key challenge is that relying solely on diagnoses results in sequences that are too short, while using only event types leads to highly repetitive sequences dominated by periodic events, such as treatment start and finish.

However, using both diagnoses and event types together introduces another issue: diagnoses are sparsely distributed within a constant stream of repeated procedures, leading to imbalanced class distributions and poor performance. To mitigate this, we filter the data by removing duplicate events between diagnoses, allowing us to retain useful treatment data while preserving class balance.

The final labels are created by converting ICD-9 and ICD-10 codes to ICD-10 chapters using General Equivalence Mapping (GEM) and adding event types as additional classes. This conversion is necessary because the number of ICD codes is too large to use them directly as classes. Additionally, we address the issue of multiple diagnoses occurring in a single event by sorting timestamps for reproducibility.

## C.5 PERFORMANCE IMPROVEMENTS

The HoTPP benchmark provides highly optimized training and inference procedures for the efficient evaluation of datasets containing up to tens of millions of events. First, we implement parallel RNN inference, which reuses computations when inference is initiated from multiple starting points within a sequence. Additionally, we optimize the code and apply PyTorch Script to the CT-LSTM model from NHP and the ODE model. CT-LSTM from EasyTPP and the ODE model from the original repository serve as baselines. The timing results are presented in Table 3. Experiments were conducted using a synthetic batch with a size of 64, a sequence length of 100, and an embedding dimension of 64. Evaluation was performed on a single Nvidia RTX 4060 GPU. The results show that HoTPP is 17 times faster at RNN autoregressive inference compared to simple prefix extension. HoTPP also accelerates CT-LSTM and ODE by 4 to 8 times. These optimizations significantly extend the applicability of the implemented methods to larger-scale datasets[8]. Additionally, HoTPP offers a GPU implementation of the Hungarian algorithm, which also finds applications in computer vision.

Table 3: HoTPP computation speed improvements in terms of seconds per batch.

| Implementation | Study | | | | |
| --- | --- | --- | --- | --- | --- |
| | RNN autoreg. inference | CT-LSTM inference | CT-LSTM train | ODE inference | ODE train |
| Baseline | 2.44 | 0.0158 | 0.0519 | 0.08 | 0.162 |
| **HoTPP** | **0.14** | **0.0025** | **0.0276** | **0.01** | **0.048** |

---

[8]Experiments are documented in the "notebooks" folder of the HoTPP repository

## D  Training details

We trained each model using the Adam optimizer, which has a learning rate 0.001, and a scheduler that reduces the learning rate by 20% after every five epochs. Gradient clipping was applied, and the maximum L2-norm was set to 1.

We performed computations on NVIDIA V100 and A100 GPUs, with some smaller experiments conducted on an Nvidia RTX 4060. Each method was trained on a single GPU. The training time varied depending on the dataset and method, ranging from 5 minutes for RMTPP on the StackOverflow dataset to 40 minutes for the same method on the Transactions dataset and up to 15 hours for NHP on Transactions. Multi-seed evaluations took approximately five times longer to complete.

## E  Hyperparameters

We analyzed the behavior of OTD and T-mAP metrics depending on their parameters. The results are shown in Figure 11. The OTD cost has little effect on the ranking of methods but significantly influences the metric scale. The choice of the $\delta$ parameter in T-mAP can, in some cases, affect method rankings, but as long as $\delta$ is not set too small or too large, the metric demonstrates stable rankings. We set the $\delta$ parameter to approximately 10-30% of the horizon duration.

Evaluation hyperparameters are presented in Table 4. Dataset-specific training parameters are listed in Table 5.

Table 4: Evaluation hyperparameters.

| Dataset | Maximum Length | OTD prefix size | OTD Cost $C$ | T-mAP horizon | T-mAP $\delta$ |
|---|---|---|---|---|---|
| Transactions | 32 | 5 | 1 | 7 (week) | 2 |
| MIMIC-IV | 32 | 5 | 2 | 28 (days) | 4 |
| Retweet | 32 | 10 | 15 | 180 (seconds) | 30 |
| Amazon | 32 | 5 | 1 | 10 (unk. unit) | 2 |
| StackOverflow | 32 | 10 | 1 | 10 (minutes) | 2 |

Table 5: Training hyperparameters.

| Dataset | Num epochs | Max Seq. Len. | Label Emb. Size | Hidden Size | Head hiddens |
|---|---|---|---|---|---|
| Transactions | 30 | 1200 | 256 | 512 | $512 \rightarrow 256$ |
| MIMIC-IV | 30 | 64 | 16 | 64 | 64 |
| Retweet | 30 | 264 | 16 | 64 | 64 |
| Amazon | 60 | 94 | 32 | 64 | 64 |
| StackOverflow | 60 | 101 | 32 | 64 | 64 |

## F  Datasets Analysis

HoTPP incorporates datasets from various domains, including finance, social networks, and healthcare. Below, we provide additional details highlighting the key differences among these datasets.

**Ordering Sensitivity.** While most datasets—such as StackOverflow, Amazon, Retweet, and Transactions—exhibit a relatively balanced distribution of timestamps, we observed that in MIMIC-IV, the proportion of zero time steps exceeds 50%, as shown in Table 6. This results in a significant number of events sharing the same timestamp. Consequently, the actual order of events within the dataset may hold greater significance during evaluation than the precise prediction of timestamps. This characteristic can lead to instability in timestamp-based metrics, such as OTD and T-mAP, compared to index-based metrics like next-event quality, pairwise MAE and accuracy. The low next-item accuracy of methods based on NHP loss (NHP, AttNHP, and ODE) on MIMIC-IV can be attributed to their independent modeling of each event class, which results in a random ordering of events with identical timestamps.

Table 6: Time step percentiles.

| Dataset | 1% | 5% | 10% | 50% | 90% | 95% | 99% |
|---|---|---|---|---|---|---|---|
| Transactions | 0.0 | 0.0 | 0.0 | 1.0 | 2.0 | 2.0 | 5.0 |
| MIMIC-IV | 0.0 | 0.0 | 0.0 | 0.0 | 6.9 | 63.8 | 768.0 |
| Retweet | 0.0 | 0.0 | 1.0 | 8.0 | 85.0 | 151.0 | 377.0 |
| Amazon | 0.01 | 0.01 | 0.01 | 0.73 | 0.79 | 0.79 | 0.80 |
| StackOverflow | 0.0003 | 0.01 | 0.05 | 0.51 | 2.16 | 2.97 | 5.08 |

**Discrete Timestamps.** Some datasets feature continuous timestamps (e.g., StackOverflow, Amazon, MIMIC-IV), while others round timestamps to a specific precision (e.g., Retweet, Transactions). Modeling discrete timestamps presents a unique challenge, as density estimation methods like NHP can produce infinite PDF values. We introduce small Gaussian noise to discrete timestamps to address this issue, effectively smoothing the distribution. The degree of smoothing was carefully tuned for each dataset individually. The exact values are provided in the configuration files included with the source code.

# G  QUALITATIVE ANALYSIS OF PREDICTIONS

Table 7 presents example predictions from various methods on the Transactions, MIMIC-IV, and Retweet datasets. The table focuses exclusively on predicted label sequences. It can be seen that all methods, except HYPRO, exhibit issues with constant or repetitive outputs. We believe this behavior stems from a bias in the predictions toward the most frequent labels, especially in scenarios with high uncertainty. For the MIMIC-IV dataset, the prediction patterns differ significantly, as most events follow a predefined sequence, such as admission, a standard set of laboratory tests, and diagnosis. In this case, the uncertainty is lower, enabling the methods to generate sequences with more diverse event types. Addressing the mentioned limitation in future research could lead to the development of methods capable of producing more varied and realistic sequences in high-uncertainty scenarios.

Table 7: Example predictions (labels only).

| Method | Seq. ID | Transactions | MIMIC-IV | Retweet |
|---|---|---|---|---|
| IFTPP | 0 | 3, 1, 3, 1, 3, 1, 3, 3, 3 | 10, 27, 23, 22, 1, 27, 3, 28, 26, 25, 23, 30 | 1, 1 |
|  | 1 | 6, 3, 6, 6, 6, 6, 6 | 11, 10 | 0, 0, 0, 0, 0, 0, 0, 0, 0 |
|  | 2 | 3, 1, 3, 1, 3, 1, 3, 1, 3 | 2, 7, 14, 12, 4, 6, 11, 10 | 1, 1, 1, 1, 1, 1, 1, 1, 1, 1, 1, 1 |
| IFTPP-K | 0 | 3, 3, 3, 3, 3, 3, 3, 3, 3, 3 | 10, 1, 1, 3, 3, 8, 9, 23, 2, 2, 5, 2 | 1, 1, 1, 1, 1, 1, 1, 1, 1, 1, 1, 1 |
|  | 1 | 6, 6, 6, 6, 6, 6, 6 | 11, 1, 3, 5, 8, 2, 9, 2, 2, 7, 7, 4 | 1, 1, 1, 1, 0, 0, 0, 0, 0 |
|  | 2 | 3, 1, 1, 1, 1, 1, 1 | 15, 4, 2, 7, 4, 1, 1, 3, 3, 2, 3, 2 | 1, 1, 1, 1, 1, 1, 1, 1, 1, 1, 1, 1 |
| RMTPP | 0 | 3, 1, 3, 1, 3, 1 | 10, 27, 23, 22, 1, 30, 27, 5, 8, 9, 16, 15 | 1, 1, 1, 1 |
|  | 1 | 6, 6, 6, 6, 6, 6 | 11, 10, 1 | 0, 0, 0, 0, 0, 0, 0, 0, 0 |
|  | 2 | 3, 1, 3, 1, 3 | 2, 7, 14, 12, 4, 6, 11, 10 | 1, 1, 1, 1, 1, 1, 1, 1, 1, 1, 1, 1 |
| RMTPP-K | 0 | 3, 3, 3, 3, 3, 3, 3 | 10, 22, 22, 28, 28, 26, 25, 23, 23, 27, 5, 8 | 1, 1, 1, 1, 1, 1 |
|  | 1 | 6, 6, 6, 6, 6, 6, 6 | 11, 1, 3 | 1, 0, 1, 0, 0, 0, 0, 0 |
|  | 2 | 3, 1, 1, 1, 1, 1 | 2, 2, 7, 7, 4, 6, 1, 3 | 1, 1, 1, 1, 1, 1, 1, 1, 1, 1, 1, 1 |
| NHP | 0 | 1, 1, 1, 1, 1, 3, 1, 1, 1 | 6 | 1, 1, 1, 1 |
|  | 1 | 1, 1, 6, 6, 6, 6, 6 | 6, 6, 6 | 1, 1, 1, 1, 1, 1, 1 |
|  | 2 | 1, 1, 1, 1, 1, 1, 1, 1, 1, 1 | 6, 2 | 1, 1, 1, 1, 1, 1, 1, 1, 1, 1, 1, 1 |
| ODE | 0 | 1, 1, 1, 1, 1, 1, 1, 1, 1, 1, 1 | 10, 1, 3, 5, 2, 4, 6, 1 | 1, 1, 1, 1 |
|  | 1 | 6, 6, 6, 6, 6 | 1, 3, 5, 2, 4, 6, 1, 3, 5, 2, 4, 6 | 0, 0, 0, 0, 0, 0, 0 |
|  | 2 | 1, 1, 1, 1, 1, 1, 1 | 2, 4, 6, 1, 3, 5, 2, 4, 10 | 1, 1, 1, 1, 1, 1, 1, 1, 1, 1, 1, 1 |
| HYPRO | 0 | 3, 1, 16, 3, 12, 1 | 23, 27, 22, 1, 28, 26, 25, 23 | 0, 0, 1, 0 |
|  | 1 | 1, 32, 6, 6, 1, 6 | 1, 28, 25, 23, 3, 26, 22, 5 | 0, 0, 0, 1, 1, 0, 1 |
|  | 2 | 3, 5, 1, 3, 5 | 15, 13, 2, 7, 4, 1, 3, 5 | 0, 0, 0, 0, 0, 0, 1, 0, 0, 0, 0, 0 |

## H  T-mAP FOR HIGHLY IRREGULAR SEQUENCES

This section analyzes a toy dataset containing highly irregular event sequences. This dataset includes a single label, which aims to predict timestamps. Most time intervals in the dataset are zero, with only 5% of intervals equal to one. We compare three simple baselines:

- **ZeroStep**, which predicts events with timestamps identical to the last observed event (zero intervals);

- **UnitStep**, which predicts events with a unit time step (the largest time step in the dataset);

- **MeanStep** predicts events using the average time step computed from historical data.

Evaluation results are shown in Figure 9. The results indicate that the MAE and OTD metrics assign the lowest error to the ZeroStep baseline, which simply repeats the last event without accounting for the dataset's irregularity. In contrast, T-mAP identifies the MeanStep baseline as the most effective, as it is the only method that analyzes historical data and incorporates timestamp statistics (mean interval) into its predictions.

These findings suggest that T-mAP is a more appropriate metric for assessing the ability of methods to predict irregular events.

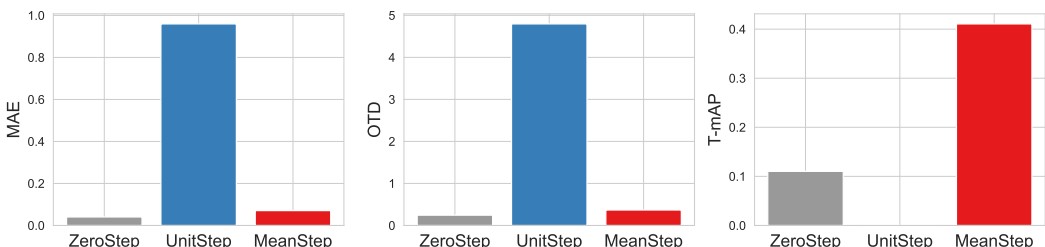

Figure 9: Comparison of simple baselines on the Toy dataset with highly irregular time intervals. MAE and OTD metrics represent error values, while T-mAP measures model quality.

## I  LONG-TAIL PREDICTION

In this section, we assess the capability of various evaluation metrics to capture long-tail prediction quality, specifically the ability of models to predict rare classes. Unlike OTD, T-mAP evaluates each class independently, allowing for different aggregation strategies. The standard T-mAP computes a macro average, where the quality for each event class contributes equally to the final score. Additionally, the HoTPP benchmark includes a weighted variant of T-mAP, where the weights are proportional to class frequencies. Figure 10 compares the performance of IFTPP, IFTPP-K, and RMTPP on the Transactions dataset, which includes 203 classes. The results show that all metrics, except macro T-mAP, remain unaffected as the dataset size increases beyond 60 classes. In contrast, macro T-mAP effectively evaluates the ability of models to predict across all available classes, highlighting its suitability for long-tail prediction tasks.

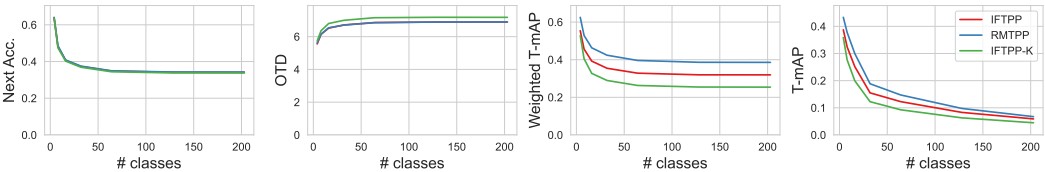

Figure 10: Comparison of various metrics on the Transactions dataset across subsets with fewer event classes.

## J    LABELS ENTROPY DEGRADATION

For simplicity, some datasets were omitted in the main part of the paper. In Figure 12, we show the dependency of label distribution entropy on step size for all datasets.

## K    THE OPTIMAL SEQUENCE LENGTH

For simplicity, some datasets were omitted from the main part of the paper. Figure 13 illustrates the relationship between label distribution entropy and step size across all datasets.

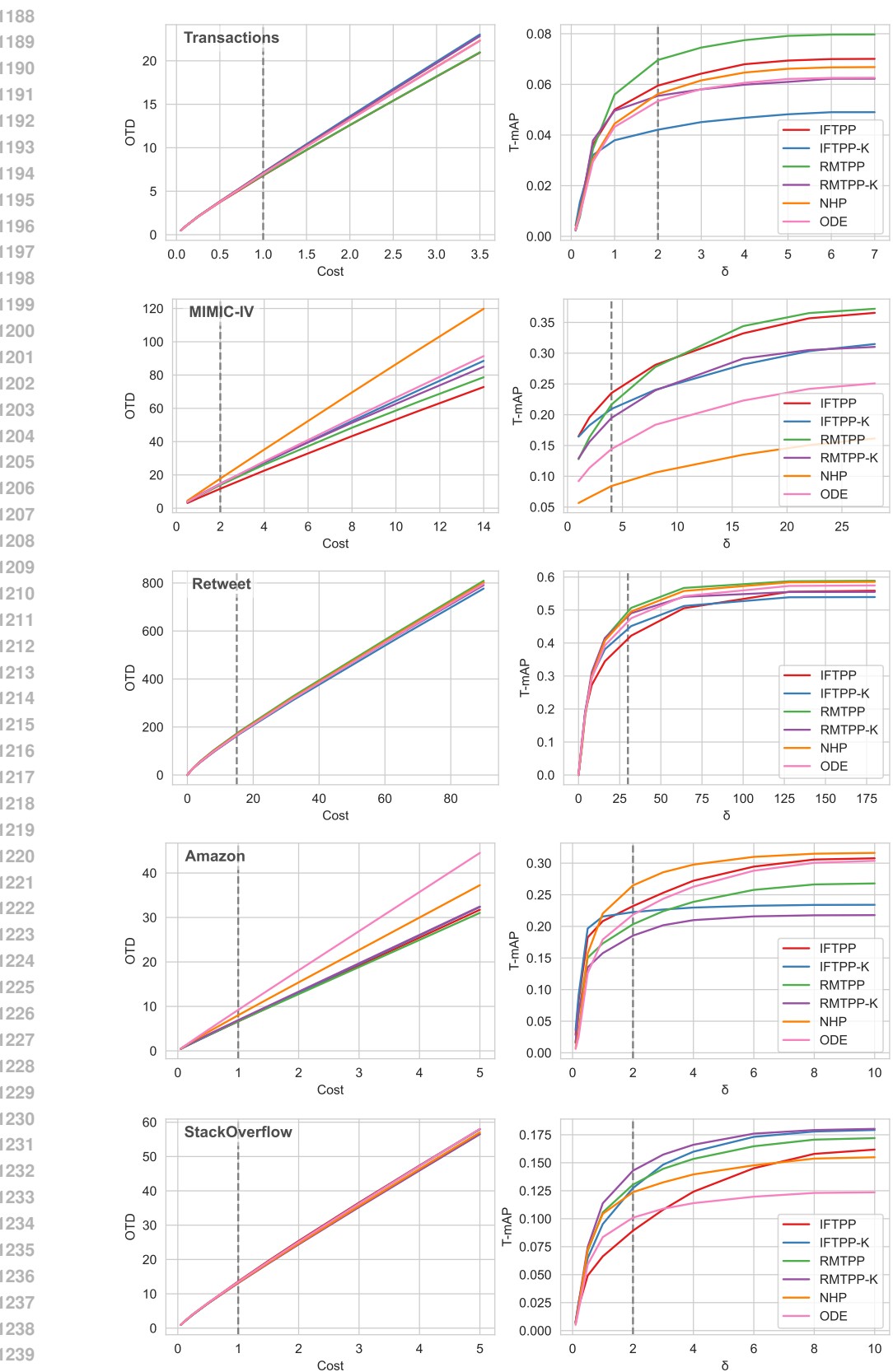

Figure 11: The dependency of metric values on the metric parameter.

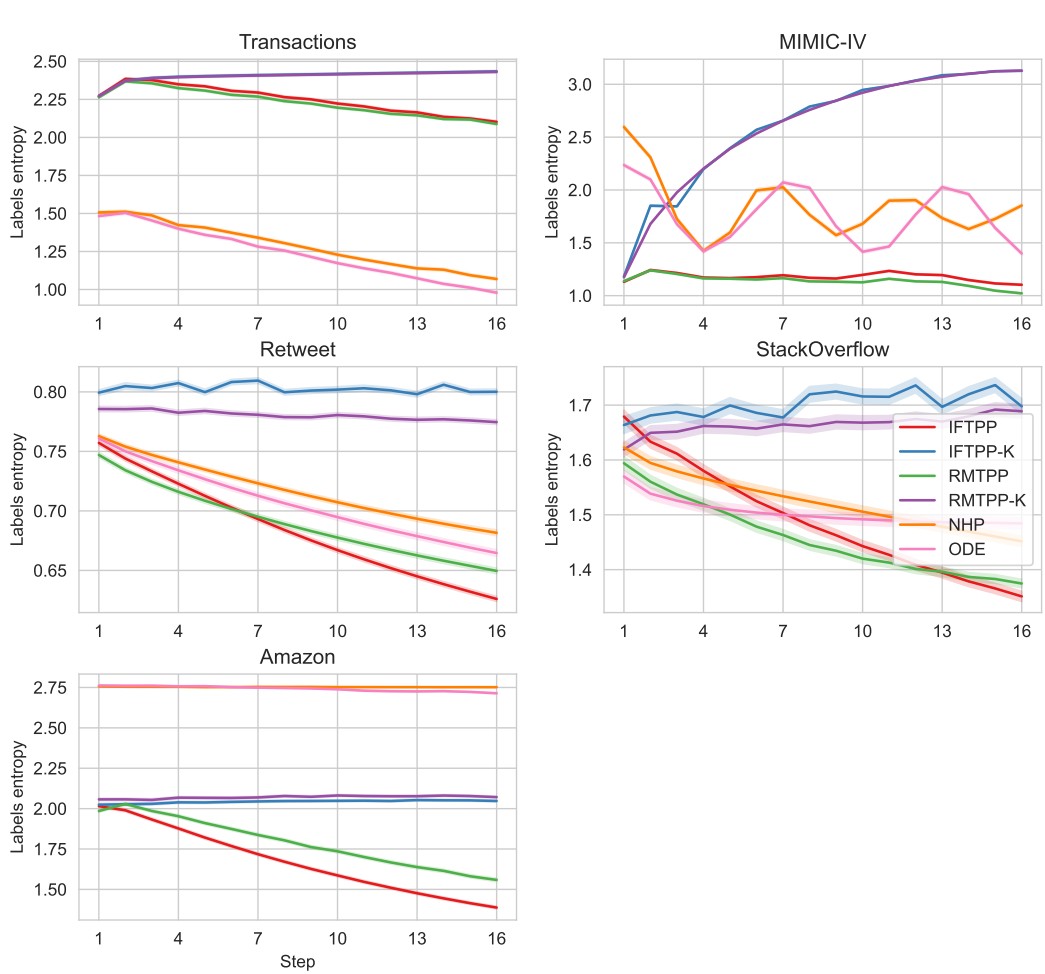

Figure 12: Entropy of label distributions as a function of the position in the generated sequence.

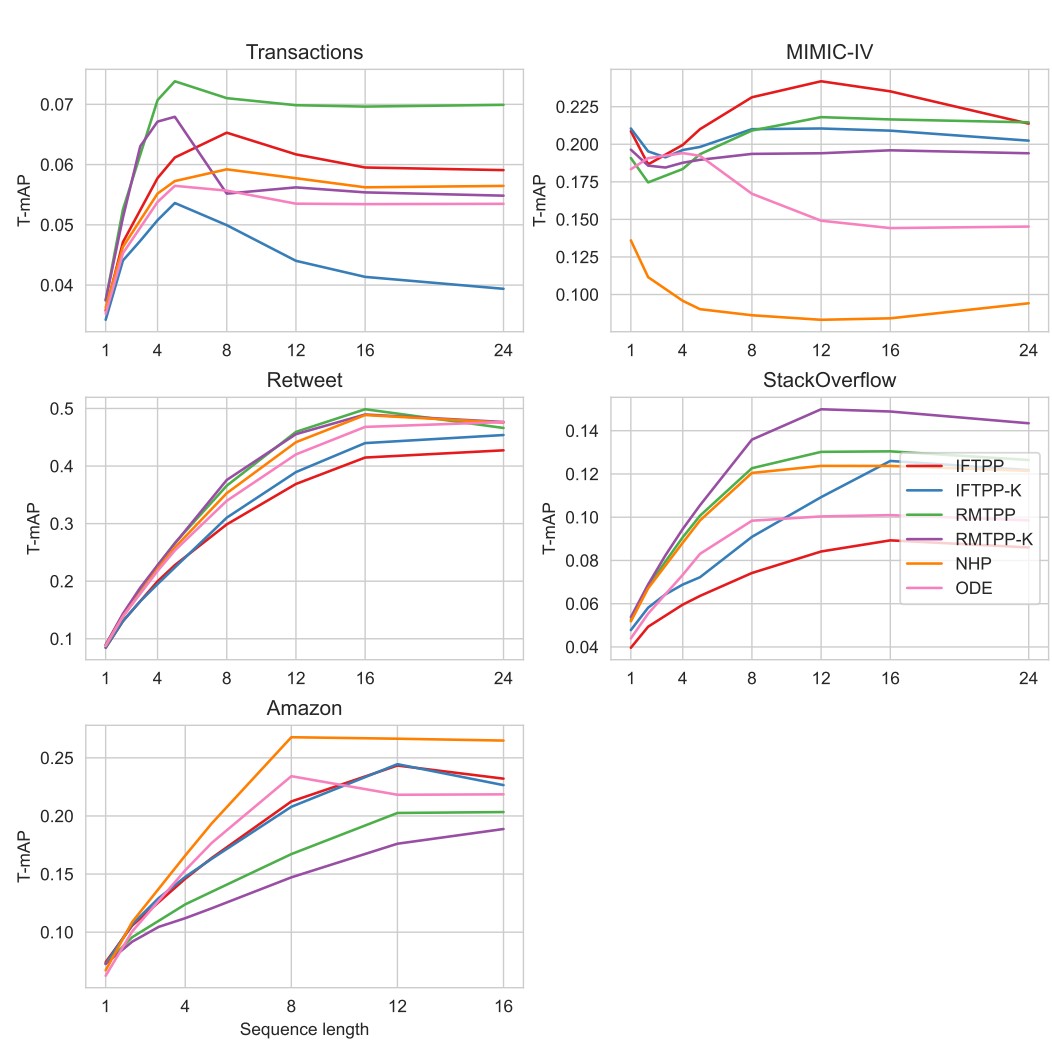

Figure 13: T-mAP dependency on the maximum length of the predicted sequence.

