# OpenReview forum: "HoTPP Benchmark: Are We Good at the Long Horizon Events Forecasting?"
_ICLR.cc/2025/Conference — Submitted to ICLR 2025_

### Official Review · Reviewer_Z31z · 2024-10-29

**Soundness:** 2
**Presentation:** 2
**Contribution:** 3
**Rating:** 5
**Confidence:** 2

**Summary:**

The authors introduce Temporal mean Average Precision (T-mAP), a temporal variant of mAP that overcomes the limitations of existing long-term evaluation metrics. They also release HoTPP, the first benchmark specifically designed to evaluate long-term MTPP predictions, which includes a large-scale dataset of up to 43 million events.

**Strengths:**

The author's motivation is clearly expressed, the approach sounds interesting, the problem of modeling long sequences of events is well addressed, and the large dataset developed has the potential to advance the field.

**Weaknesses:**

1. The paper is difficult to understand. What is the ‘structured descriptions of each event’, what is the challenges of autoregressive prediction in line 48. What are the challenges of autoregressive prediction in line 052? How are subsequences handled for each event type in line 99. This increases the difficulty for readers to understand, and it is recommended that the author explain these terms in more detail.

2. The motivation is vague. Why do we need long-term event prediction? In fact, we can execute Next-event task multiple times to get similar results.

3. How does the proposed metrics perform in a long-tail prediction scenario?

4. Does the proposed metric take into account the time error of event occurrence? If the time interval corresponding to the event is far away, can the method cope with this situation?

**Questions:**

See Weaknesses.

---

> ### Author Response · Authors · 2024-11-26
>
> We sincerely thank you for your thoughtful and comprehensive feedback. Your insights and questions have provided us with an opportunity to clarify and further strengthen our work. Below, we address each of your points in detail, highlighting the revisions and improvements made in response to your suggestions.
>
> **W1.a Structured descriptions of each event.** Thank you for highlighting this ambiguity. We have clarified this point in the updated version of the paper. Specifically, the field of Marked Temporal Point Processes (MTPP) pertains to tabular data processing. Tabular data is structured, meaning it consists of multiple columns of different types [1].
>
> **W1.b Challenges of autoregressive prediction.** The mentioned sentence directly follows a discussion outlining one of these challenges, specifically the dependency of autoregressive methods on their own prediction errors. We identify and analyze the challenges associated with autoregression in Section 6.3 (Autoregression vs direct horizon prediction).
>
> **W1.c How subsequences are handled (line 99).** As noted in the text, details on MTPP modeling are provided in Appendix A. Modeling MTPPs involves a deep understanding of probability distribution parameterization techniques and sampling. Rizoiu [2] offers an excellent introduction to these concepts. We chose to include these details in the Appendix because they can be complex to grasp initially and are not essential for understanding the core contributions of our work.
>
> **W2. The motivation for the long-term event prediction?** The paper provides both theoretical and empirical justification for the importance of long-horizon prediction and evaluation. First, achieving high next-event prediction quality does not necessarily result in good long-horizon predictions. In Section 6.1, we demonstrate that HYPRO, a method specifically designed for long-horizon prediction, often performs worse in terms of next-event quality. This highlights the need for distinct evaluation approaches for different prediction horizons. The theoretical foundations are presented in Section 3. Notably, next-event quality assessment is analogous to OTD with a prefix size of 1, which leads to incorrect accounting for false positives and false negatives. This issue becomes especially important when a model predicts too many or too few events over the horizon.
>
> **W3. Long-tail prediction.** T-mAP employs macro averaging of individual scores for all classes, ensuring that the prediction quality of each class contributes equally to the final metric. This makes T-mAP particularly effective for studying long-tail prediction, in contrast to metrics like error rate and OTD, which may overlook performance on less frequent classes. To illustrate this capability, we added Appendix I, where we demonstrate how T-mAP effectively evaluates long-tail prediction on the Transactions dataset, which includes a total of 203 classes.
>
> **W4. Measuring time quality.** Time prediction is a core challenge in the Marked Temporal Point Processes (MTPP) domain. All metrics, including next-event MAE, OTD, and T-mAP, evaluate predicted timestamps. In Section 3, we highlight the advantages of using T-mAP for time prediction assessment. Specifically, we show that T-mAP accurately accounts for false positives and false negatives, a key property lacking in OTD. To address the challenge of irregular timestamps, we included an additional example in Appendix H. In this example, we generated a toy dataset with highly irregular timestamps and demonstrated that both next-event MAE and OTD assign the lowest error to a simple baseline that repeats the last observed event. In contrast, T-mAP favors a baseline that captures time step statistics. This result suggests that T-mAP is better suited for evaluating model performance in scenarios with high irregularity.
>
> Thank you once again for your thoughtful feedback. Please let us know if we have adequately addressed your questions. If so, we would greatly appreciate it if you could consider updating your final score for the paper.
>
> [1] Ryan M. Deep learning with structured data. – Simon and Schuster, 2020.
>
> [2] Rizoiu M. A. et al. Hawkes processes for events in social media //Frontiers of multimedia research. – 2017. – С. 191-218.

---

> ### Author Response · Authors · 2024-11-29
> **Discussion**
>
> As the discussion period is coming to a close, we kindly ask if we have adequately addressed all your questions and concerns. If our responses are satisfactory, we would greatly appreciate it if you could consider adjusting the final score accordingly. Thank you for your time and thoughtful feedback!

---

### Official Review · Reviewer_nKzJ · 2024-11-01

**Soundness:** 2
**Presentation:** 2
**Contribution:** 2
**Rating:** 3
**Confidence:** 2

**Summary:**

This paper benchmarks the task of long horizon events forecasting. The main contribution of this paper is to propose a new metric, namely T-mAP. Moreover, this paper also includes large-scale datasets with up to 43 million events. Although with these contributions, this paper is not well-written and not easy to understand.

**Strengths:**

- This paper proposes a new metric, namely T-mAP, to better evaluate the performance of long horizon events forecasting.

- This paper includes large-scale datasets with up to 43 million events.

- This paper release HoTPP open-source benchmark to facilitate future research.

**Weaknesses:**

- This paper is not well-written and hard to follow. The main contribution of this paper is to propose T-mAP to evaluate models on the task of long horizon events forecasting. However, why T-mAP is better than the existing metric, i.e., OTD, is not clearly introduced.

- The experiments are not extensive. In the title of this paper is about long-horizon event forecasting, but the experiments covers equally on the topic of the next-item forecasting.

- The number of compared baselines is not many, and the baselines are not proposed recently. It seems that the topic of event forecasting is not very hot, so the motivation that we need a benchmark is very strong.

**Questions:**

- How T-mAP is better than the existing metric?

- More analysis on long-horizon event forecasting are needed.

- Why the baselines are not new?

---

> ### Author Response · Authors · 2024-11-26
>
> We deeply appreciate your detailed review and the time you have dedicated to evaluating our work. In the responses below, we address your comments point by point, highlighting the changes made in the updated version of the manuscript and providing additional context where needed.
>
> **W1 / Q1. Why T-mAP is better than OTD.** We discuss the limitations of OTD and the benefits of T-mAP in Section 3. There are two main issues with OTD. First, previous works used OTD with fixed-length prefixes. This approach incorrectly counts false positives and false negatives. T-mAP, instead, uses time horizons to compare all events within a specific interval. This ensures accurate counting and is especially useful for models that generate events too rarely or too frequently. Second, OTD relies on hard labels and needs calibrated model outputs to reduce errors. This calibration step was often missing in earlier studies. T-mAP does not have this issue because it is invariant to linear calibration. This makes T-mAP more robust and reliable for comparisons. In the updated version of the paper, we also included Appendix H, which illustrates T-mAP's ability to evaluate highly irregular event sequences, and Appendix I, which highlights its importance in assessing long-tail prediction, to further highlight the benefits of the proposed metric.
>
> **W2 / Q2. Experiments covers equally on the topic of the next-item forecasting.** The main focus of our work is developing an evaluation procedure for long-horizon prediction. To highlight its importance, we demonstrate how it differs from prior approaches like next-event prediction and OTD. Analyzing metrics such as next-event MAE and error rate is essential to this comparison. Additionally, reporting widely-used metrics improves the benchmark's quality, supports reproducibility, and enables meaningful comparisons in future.
>
> **W3 / Q3. Domain importance.** For a discussion on the domain of event sequences, please refer to the general response (Domain and benchmark importance).
>
> **W3 / Q3. Baselines.** Our experimental evaluation includes a wide range of popular baselines that represent diverse methodologies. These include RNNs and Transformers, intensity-based and intensity-free approaches, next-event and horizon prediction methods, discrete and continuous-time models, including ODE-based approaches. Unfortunately, many recent works face challenges with formulation, reproducibility, or scale. For instance, the implementation of ContiFormer [1] is limited to a toy example (a spiral dataset) which requires over 2 hours of training on an RTX 4060 GPU, questioning the actual effectiveness of the approach. Furthermore, the authors have not addressed or commented on reproducibility concerns raised on GitHub. In contrast, we provide implementation, hyperparameters, and full evaluation results for ALL methods considered, achieving exceptional reproducibility in the event sequence domain.
>
> Thank you once again for your thoughtful feedback. Please let us know if we have adequately addressed your questions. If so, we would greatly appreciate it if you could consider updating your final score for the paper.
>
> [1] Chen Y. et al. “Contiformer: Continuous-time transformer for irregular time series modeling”, NeurIPS, 2024

---

> ### Author Response · Authors · 2024-11-29
> **Discussion**
>
> As the discussion period is coming to a close, we kindly ask if we have adequately addressed all your questions and concerns. If our responses are satisfactory, we would greatly appreciate it if you could consider adjusting the final score accordingly. Thank you for your time and thoughtful feedback!

---

### Official Review · Reviewer_adKC · 2024-11-03

**Soundness:** 2
**Presentation:** 2
**Contribution:** 2
**Rating:** 3
**Confidence:** 2

**Summary:**

The paper introduces a new metric, Temporal mean Average Precision (T-mAP), which is designed for evaluating long-horizon predictions in marked temporal point processes (MTPP). This offers a new perspective on evaluating forecasting accuracy beyond traditional metrics like OTD. It also introduce HoTPP, a new benchmark for long-horizon forecasting, which provides large datasets and optimized procedures for both autoregressive and parallel inference.

**Strengths:**

1. The paper is well-organized, with clear sections delineating the introduction, related work, methodology, experiments, and conclusions.
2. Figures and tables are used effectively to illustrate concepts and present experimental results.
3. The significance of forecasting multiple future events is substantial.

**Weaknesses:**

1. The motivation in section 3, while intended to address deficiencies in existing methods such as OTD, appears weakly articulated. The example provided in Figure 2 lacks clarity, particularly why OTD cannot correctly match the last triangle with the ground truth. Furthermore, the assertion that OTD is computed between fixed-size prefixes is confusing since n_p and n_gt vary, which contradicts the description in line 178 to 185.
2. Some claims are not clear. For example, in line 230 T-mAP identifies the matching that maximizes both precision and recall simultaneously. Typically, these metrics are in a trade-off relationship. How T-mAP manages to maximize both?
3. The experimental section (Section 6.1) lacks specific references to the methods and datasets discussed, which makes it difficult to follow and verify the stated findings. Direct references to specific methods and datasets should be included to enhance clarity.

**Questions:**

1. Deep methods do not necessarily outperform rule-based methods, how to illustrate the superiority of the proposed metric by comparing them?
2. What is the main message of section 4.3? It seems to only show that the metric increases monotonically with delta.
3. Can authors discuss which scenarios are most suitable for different types of metrics? Like how does the proposed metric handle sparse or highly irregular data compared to traditional ones?

---

> ### Author Response · Authors · 2024-11-26
>
> We sincerely thank you for your detailed and constructive feedback. Below, we address each of your points thoroughly, referencing the corresponding updates made in the revised text.
>
> **W1. T-mAP motivation.** Optimal Transport Distance (OTD) can indeed be computed between sequences of varying lengths. However, previous works have only utilized fixed-size prefixes [1, 2], which lead to the challenges discussed in Section 3, particularly illustrated in Figure 2.1. One of our key contributions is addressing time horizons rather than relying on prefixes of predefined lengths. While it is theoretically possible to implement a form of "horizon" OTD, such a metric would only address one of the two issues outlined in Section 3. The second issue, related to model calibration, requires the use of label distributions rather than hard predictions. Based on these considerations, we conclude that T-mAP is a superior alternative to both OTD and "horizon" OTD. Additionally, T-mAP offers more intuitive hyperparameters, such as horizon length and maximum allowable time error, compared to the abstract insertion and deletion costs associated with OTD. In the updated version of the paper, we included Appendix H, which illustrates T-mAP's ability to evaluate highly irregular event sequences, and Appendix I, which highlights its importance in assessing long-tail prediction, to further highlight the benefits of the proposed metric.
>
> **W2. Precision & Recall.** Indeed, precision and recall are inherently in a trade-off relationship when varying the decision threshold, as thresholding directly controls the balance between true positives and false positives. However, this trade-off does not apply during the matching process, where the goal is to establish an optimal alignment between predictions and ground truth events. In matching, precision and recall grow together because the alignment maximizes the number of true positives (i.e., correctly matched pairs). Importantly, the total number of predictions and ground truth events remains unchanged, as these are independent of the alignment process. Consequently, the denominators in the precision and recall formulas are constant, allowing both metrics to be optimized simultaneously through matching.
>
> **W3. References to datasets and methods in Section 6.1.** We introduce both datasets and methods at the start of the Experiments section (Section 6). This structure allows us to maintain a clean and concise presentation for individual experiments, including subsection 6.1. All the methods considered are implemented in our source code, eliminating the need for additional references. Exact links to the datasets are provided in Appendix C.4, and our source code includes data preparation scripts for all datasets (located in the experiments folder). All datasets, except MIMIC-IV, are downloaded automatically. Due to licensing constraints, MIMIC-IV must be obtained individually by each user.
>
> **Q1. Deep learning suboptimality and the proposed T-mAP metric.** We address this question at the beginning of Section 6.1. Unlike most machine learning models, rule-based methods do not predict label distributions; instead, they produce hard labels. This limitation prevents rule-based methods from being adjusted to specific precision or recall levels. In our experiments, we demonstrate that T-mAP, unlike OTD, effectively captures this distinction between baselines and deep learning models.
>
> **Q2. Message of section 4.3.** The primary objective of this section is to provide guidance on selecting hyperparameters for the T-mAP metric. We conclude that the T-mAP delta parameter can be set to either twice the cost of OTD or just beyond the initial slope of the delta-quality curve, as smaller delta values impose overly strict constraints on time prediction accuracy. Additionally, we demonstrate that T-mAP is robust to the choice of delta, with small variations in this parameter having minimal impact on the evaluation procedure.
>
> **Q3. Sparse and highly irregular data.** In the MTPP domain, each event is represented as a point in time. The sparsity of events can be adjusted by simply altering the unit of time measurement. To address irregularity, we have included an additional example in Appendix H. In this example, we generate a toy dataset with highly irregular timestamps and show that both next-event MAE and OTD assign the lowest error to a simple baseline that merely repeats the last observed event. In contrast, T-mAP favors a baseline that learns the average time interval. Based on this analysis, we conclude that T-mAP provides a more accurate assessment of model performance in scenarios characterized by high irregularity.
>
> Please let us know if we have adequately addressed your questions.
>
> [1] Xue S. et al. “EasyTPP: Towards Open Benchmarking Temporal Point Processes”, ICLR 2024
>
> [2] Xue S. et al. “Hypro: A hybridly normalized probabilistic model for long-horizon prediction of event sequences”, NeurIPS 2022

---

> ### Author Response · Authors · 2024-11-29
> **Discussion**
>
> As the discussion period is coming to a close, we kindly ask if we have adequately addressed all your questions and concerns. If our responses are satisfactory, we would greatly appreciate it if you could consider adjusting the final score accordingly. Thank you for your time and thoughtful feedback!

---

### Official Review · Reviewer_X3UT · 2024-11-04

**Soundness:** 3
**Presentation:** 2
**Contribution:** 2
**Rating:** 5
**Confidence:** 3

**Summary:**

This paper introduces HoTPP (Horizon Temporal Point Process Benchmark), a benchmark for evaluating long-horizon event sequence prediction models. Its main contributions include: proposing a novel evaluation metric called Temporal mean Average Precision (T-mAP), inspired by object detection metrics in computer vision, which properly handles variable-length sequences within a prediction horizon and accounts for false positives and false negatives; demonstrating through comprehensive analysis that models with high next-event prediction accuracy don't necessarily perform well at long-horizon forecasting, suggesting the need for specialized approaches for each task; developing the HoTPP benchmark, which includes large-scale datasets from diverse domains (finance, healthcare, social networks) with up to 43 million events, implementation of various modeling approaches (including rule-based baselines, intensity-free methods, intensity-based methods, and Next-K prediction models), optimized procedures for both autoregressive and parallel inference, and theoretical proof of T-mAP computation correctness; and revealing through extensive experiments across multiple datasets the trade-offs between next-event and long-horizon prediction performance, benefits of Next-K approaches for long-horizon predictions, importance of proper sequence length selection, and analysis of label distribution entropy degradation in autoregressive predictions.

**Strengths:**

1. Innovative Evaluation Metric: The paper introduces Temporal mean Average Precision (T-mAP), a novel metric inspired by object detection, which addresses limitations in existing evaluation methods for long-horizon forecasting by accurately handling false positives and negatives, offering a refined measurement of model performance in Marked Temporal Point Processes (MTPP).
2. Practical Benchmark: The HoTPP benchmark developed in this work includes large-scale, diverse datasets and optimized inference procedures, establishing a standardized framework that supports both autoregressive and parallel inference, greatly enhancing research accessibility and reproducibility in long-horizon event forecasting.
3. Comprehensive Empirical Analysis: The paper rigorously evaluates various models, including rule-based and advanced neural methods, across multiple datasets, providing robust empirical evidence that reveals critical insights into the performance trade-offs of next-event vs. long-horizon forecasting.
4. Clear and Structured Presentation: The paper clearly articulates the challenges in long-horizon event prediction, explaining the proposed methodology and its advantages with illustrative figures and well-organized tables, making complex concepts acc

**Weaknesses:**

1. Limited Justification of Metric Selection: While the T-mAP metric is an innovative contribution, the paper could strengthen its justification for why T-mAP is superior to other established metrics in specific application scenarios. Including more detailed comparisons with alternative metrics, such as Order-preserving Wasserstein Distance (OPW), could provide further evidence of T-mAP's effectiveness, especially in complex event sequences.
2. Lack of Fine-Grained Analysis for Different Domains: The datasets span diverse fields like healthcare and social media. However, the paper does not profoundly explore how T-mAP performs within these domains. Analyzing domain-specific challenges or model performance variations across fields could add depth and highlight the metric’s adaptability, further demonstrating HoTPP’s real-world applicability.
3. Computational Constraints on Benchmark Implementation: Some models, like the continuous-time LSTM, require extensive computational resources, limiting their practical applicability. The paper could improve by suggesting or including optimizations, such as more efficient GPU implementations or leveraging hybrid models, making the benchmark more accessible to researchers with limited resources.
4. Limited Exploration of Next-K Models: Although the paper discusses Next-K models and their potential in improving long-horizon forecasting, there is little exploration of variations within this model family. Providing examples or implementing alternative Next-K structures could substantiate the claims regarding their advantages, offering actionable insights for researchers interested in non-autoregressive alternatives.
5. Lack of Qualitative Error Analysis: The paper could benefit from qualitative error analysis to clarify why some models underperform on long-horizon metrics. Visual examples or error case studies might offer valuable insights into prediction failures, guiding future model improvements by highlighting common error patterns in long-horizon event forecasting.

**Questions:**

1. Could the authors provide more detailed reasoning on why T-mAP is the most suitable metric for long-horizon MTPP evaluation? A comparison with other metrics, such as OPW, would help clarify the unique advantages of T-mAP for certain datasets or applications. Are there specific scenarios where T-mAP particularly excels?

2. How does T-mAP adapt to different domains represented in the HoTPP benchmark, such as healthcare vs. social media? It would be helpful if the authors could provide additional domain-specific analysis or clarify if they observed any notable trends in metric performance across different datasets.

4. Given the computational intensity of some methods (e.g., continuous-time LSTM), what optimizations, if any, do the authors recommend for users with limited hardware resources? Would simplifying certain models or using hybrid methods maintain benchmark validity while improving accessibility?

4. Next-K models are briefly discussed, but can the authors elaborate on alternative structures or settings within this family of models? Exploring how these models perform differently across long-horizon tasks could provide insights into their benefits or limitations and would help clarify if more complex Next-K structures could outperform standard autoregressive models.

---

> ### Author Response · Authors · 2024-11-26
>
> We thank you for your thoughtful and detailed feedback. Your comments have provided valuable insights and have given us the opportunity to clarify and strengthen our work. Below, we address each of your points thoroughly, referencing updates made in the revised paper and providing additional explanations where necessary.
>
> **W1/Q1. Limited Justification of Metric Selection.** Metrics are a fundamental component of experimental design and cannot be easily validated solely through experimental results. One potential approach is to link the developed metric to human evaluation, effectively providing a proxy for a complex evaluation process. However, in the domain of TPPs, there is no well-established procedure for human evaluation, and even defining a straightforward target for such assessments is challenging. As a result, we justify the proposed metric based on its intrinsic properties, which can be demonstrated both theoretically and through toy examples, as provided in Section 3. In the updated version of the paper, we included Appendix H, which illustrates T-mAP's ability to evaluate highly irregular event sequences, and Appendix I, which highlights its importance in assessing long-tail prediction, to further highlight the benefits of the proposed metric.
>
> **W1/Q1. Order-Preserving Wasserstein Distance (OPW).** Regarding the OPW metric, it is important to note that it is based on event indices and does not account for timestamps. Since timestamp modeling is a critical objective in the TPP field, we excluded OPW from our comparisons.
>
> **W2/Q2. Lack of Fine-Grained Analysis for Different Domains.** In the revised version of the paper, we included a comparison across different domains in Appendix F. We demonstrate that T-mAP is a suitable evaluation measure in domains where accurate timestamp forecasting is more critical than precise event ordering. However, in cases where datasets contain a large proportion of zero time steps and the ordering of events with identical timestamps becomes significant, T-mAP may miss some important aspects. Nevertheless, we believe that ordering events with identical timestamps should generally be avoided in practical applications.
>
> **W3/Q3. Computational Constraints.** We discuss the implemented optimizations in Section 5 (computational effectiveness) and Appendix C.5. HoTPP introduces highly efficient parallel inference procedures, as well as optimized implementations for NHP and ODE. These enhancements achieve up to a 17x improvement in performance compared to previous implementations, which is crucial for handling larger datasets. To the best of our knowledge, this is the first time approaches like NHP and ODE have been evaluated on large datasets such as MIMIC-IV and Transactions. For scenarios with limited computational resources, we recommend using mixed-precision training, which reduces memory consumption and improves training speed.
>
> **W4/Q4. Limited Exploration of Next-K Models.** Next-K models, such as those introduced in our paper (e.g., IFTPP-K and RMTPP-K), have not been previously applied in the domain of MTPPs. As a result, our work establishes a baseline for the future development of these methods. In Section 6.3, we highlight the potential benefits of this class of models. However, the further advancement of Next-K approaches lies beyond the scope of this work, which focuses on establishing a benchmark and validation procedure for the long-horizon prediction problem.
>
> **W5/Q4. Lack of Qualitative Error Analysis.** We included a qualitative analysis and a discussion of common issues in Appendix G. As observed, most methods tend to produce constant or repetitive patterns, whereas rescoring with HYPRO generates more diverse and natural outputs. We believe these challenges can be addressed in future research. Although these results are not directly related to our core contributions—namely, establishing a benchmark and metric—we have chosen to include them in the appendix for completeness.
>
> Thank you once again for your feedback. Please let us know if we have adequately addressed your questions. If so, we would greatly appreciate it if you could consider updating your final score for the paper.

---

> ### Author Response · Authors · 2024-11-29
> **Discussion**
>
> As the discussion period is coming to a close, we kindly ask if we have adequately addressed all your questions and concerns. If our responses are satisfactory, we would greatly appreciate it if you could consider adjusting the final score accordingly. Thank you for your time and thoughtful feedback!

---

### Author Response · Authors · 2024-11-26

We sincerely thank the reviewers for spending time and providing detailed feedback. We recognize that the main concerns focus on the proposed T-mAP metric. We address these concerns with additional empirical results and clarifications. We also want to emphasize other contributions of our work, which we think were not properly recognized.

**Theoretical Contributions:**

1. We identify and address the limitations of the OTD and next-event metrics in the MTPP domain, namely incorrect accounting for false positives and false negatives, and sensitivity to calibration (Section 3). The latter is important limitation, as calibration is omitted in previous works [1, 2, 3].
2. We introduce the T-mAP metric, which resolves these limitations (Section 4).
3. We prove matching consistency among decision thresholds in the T-mAP computation algorithm, a type of proof that is notably absent in computer vision [4]. Additionally, we demonstrate the invariance of T-mAP to linear calibration (Appendix B).

**Technical Contributions:**

1. We implemented parallel inference starting from multiple points within the input sequence, which is crucial for long-horizon evaluation. In Appendix C.5 we demonstrate, that our parallel autoregressive RMTPP inference on StackOverflow is 12 times faster than multiple inference calls with different prefix sizes.
2. We optimized continuous-time models using PyTorch JIT, achieving substantial speed improvements. As shown in Appendix C.5, our CT-LSTM is twice as fast as EasyTPP, and our ODE-RNN is 4 times faster than the original implementation \[3\].
3. We provided a GPU-accelerated linear sum assignment solver, that attracted the attention of computer vision researchers. In CV, this algorithm is typically executed on the CPU. Our GPU implementation is about 16 times faster than CPU in our experiments.

**Empirical Contributions:**

1. We for the first time apply advanced methods like NHP and ODE-RNN [3] to large datasets such as MIMIC-IV and Transactions. The latter is 30 times larger than previously studied datasets, like Retweet. Large datasets present new challenges in computational efficiency, which we addressed above. Dataset statistics are described in Appendix C.4.
2. We evaluate Next-K approaches for MTPP for the first time, showing that they are competitive with autoregressive methods while offering higher computational efficiency. For example, on the StackOverflow dataset, sequence generation with IFTPP-K is four times faster compared to IFTPP.

**T-mAP motivation**

We provide a theoretical foundation for T-mAP in Section 3, demonstrating that it resolves issues in accounting for false positives and false negatives inherent in next-event and OTD metrics. Additionally, we show that, unlike label error rate and OTD, T-mAP is invariant to linear calibration—a crucial property given that most prior works do not incorporate calibration. Our experiments further reveal that next-event and OTD metrics fail to capture model confidence effectively, often favoring simplistic copy-and-paste baselines. These metrics also struggle to reflect the long-horizon prediction improvements achieved by HYPRO. Furthermore, we added Appendix H, which illustrates T-mAP's ability to evaluate highly irregular event sequences, and Appendix I, which highlights its importance in assessing long-tail prediction. Together, these findings underscore our central claim that T-mAP is a more robust and sensitive metric for evaluating long-horizon predictions compared to existing alternatives.

**Domain and benchmark importance**

Interest in event sequence modeling continues to grow [1], yet the field remains underexplored, largely due to the sensitive nature of the data involved. Event sequences are common in domains such as finance (e.g., banking), retail, and healthcare, where strict data privacy requirements often necessitate extensive anonymization. This limits the availability of datasets for research and hinders progress in the field. Despite these challenges, modeling event sequences is essential for these industries. Accurate models enable critical applications, including strategic planning, personalized communication, and advanced analytics. These capabilities provide significant benefits, such as financial gains in finance and retail or improved diagnostic accuracy in healthcare. To address these needs, our benchmark sets a new standard by emphasizing speed, scalability, reproducibility, and a diverse range of evaluation tasks, being the first benchmark for long-horizon prediction in the field.

[1] Xue S. et al. “EasyTPP: Towards Open Benchmarking Temporal Point Processes”, ICLR 2024

[2] Xue S. et al. “Hypro: A hybridly normalized probabilistic model for long-horizon prediction of event sequences”, NeurIPS 2022

[3] Rubanova Y. et al. “Latent ordinary differential equations for irregularly-sampled time series” NeurIPS 2019

[4] Lin T. Y. et al. “Microsoft CoCo: Common objects in context”, ECCV 2014

---

### Meta-Review · Area_Chair_9tN6 · 2024-12-10

**Metareview:**

The reviewers find the paper difficult to understand and raise many general questions that cannot be resolved in a rebuttal as most of them  require substantial changes to the original submission and in sum a major revision. For example, it does not become obvious why the solved problem is challenging and hence, why it is important? Are there use cases that would exemplify the need for long-term predictions? When the basic understanding of the problem setting is not given, it is almost impossible to understand the contribution. Reviewers also highlighted many other issues in their reviews, eg. regarding the handling of subsequences, empirical evaluation, etc. In sum, at the current stage, the paper is not yet ready for publication at ICLR.

**Additional Comments On Reviewer Discussion:**

Unfortunately, there was no communication between authors and reviewers. Although the authors tried to provide answers and clarifications to the reviewer questions, none of them got back to the authors. However, we have to admit that the paper is difficult to comprehend and likely needs a major revision before content can be truly assessed.

---

### Decision · Program_Chairs · 2025-01-22

Reject